# CTBench: Cryptocurrency Time Series Generation Benchmark

**Yihao Ang**[1]   **Qiang Wang**[1]   **Qiang Huang**[2,*]   **Yifan Bao**[1]   **Xinyu Xi**[1]
**Anthony K. H. Tung**[1]   **Chen Jin**[1]   **Zhiyong Huang**[1]

[1]Department of Computer Science, National University of Singapore
[2]School of Intelligence Science and Engineering, Harbin Institute of Technology (Shenzhen)
`{yihao_ang, yifan_bao, atung, huangzy}@comp.nus.edu.sg`
`{qwang, xinyu_xi}@u.nus.edu  disjinc@nus.edu.sg`
`huangqiang@hit.edu.cn`

## Abstract

Synthetic time series are vital for data augmentation, stress testing, and prototyping in quantitative finance. Yet in cryptocurrency markets, characterized by 24/7 trading, extreme volatility, and rapid regime shifts, existing Time Series Generation (TSG) methods and benchmarks often fall short, jeopardizing practical utility. Most prior work targets non-financial or traditional financial domains, focuses narrowly on classification and forecasting while neglecting crypto-specific complexities, and lacks critical financial evaluations, particularly for trading applications. To bridge these gaps, we introduce **CTBench**, the first **C**ryptocurrency **T**ime series generation **Bench**mark. It curates an open-source dataset of 452 tokens and evaluates models across 13 metrics spanning forecasting accuracy, rank fidelity, trading performance, risk assessment, and computational efficiency. A key innovation is a dual-task evaluation framework: the Predictive Utility measures how well synthetic data preserves temporal and cross-sectional patterns for forecasting, while the Statistical Arbitrage assesses whether reconstructed series support mean-reverting signals for trading. We systematically benchmark eight state-of-the-art models from five TSG families across four market regimes, revealing trade-offs between statistical quality and real-world profitability. Notably, CTBench provides ranking analysis and practical guidance for deploying TSG models in crypto analytics and trading applications. The source code is available at `https://github.com/MilleXi/CTBench/`.

## 1 Introduction

Time Series Generation (TSG) has become foundational for numerous downstream tasks, including data augmentation (Bao et al., 2024; Ramponi et al., 2018), anomaly detection (Ang et al., 2023b; Wang et al., 2021), privacy preservation (Jordon et al., 2018; Tian et al., 2024), and domain adaptation (Cai et al., 2021; Li et al., 2022b). The core objective of TSG is to synthesize sequences that preserve the temporal dependencies and structural characteristics of real-world data. Despite growing interest, the vast majority of existing TSG benchmarks and methods target domains such as healthcare, mobility, or sensor data (Ang et al., 2023a; 2024). Financial time series, which are inherently noisy, non-stationary, and adversarial, remain underexplored in the context of generative modeling. More importantly, even financial TSG efforts primarily focus on stock data (Yoon et al., 2019; Wiese et al., 2020), often under simplifying assumptions that fail to generalize to emerging financial modalities. Consequently, the unique characteristics of modern financial markets, particularly in the digital asset space, are largely overlooked.

**Cryptocurrencies**, as a prominent subclass of financial time series with a global market capitalization exceeding \$4 trillion as of May 2025 (Reuters, 2025), introduce new modeling and evaluation challenges. Unlike traditional financial instruments, crypto markets operate 24/7, lack intrinsic valuation anchors, and exhibit extreme volatility driven by speculation, fragmented liquidity,

---

*Corresponding author.

and decentralized exchange infrastructure. These properties violate assumptions embedded in existing financial benchmarks (Hu et al., 2025; Wang et al., 2025; Qiu et al., 2024), which typically rely on regular trading hours, stable macroeconomic signals, or broad stationarity assumptions.

While recent benchmarks for financial time series, such as FinTSB (Hu et al., 2025) and FinTSBridge (Wang et al., 2025), have advanced evaluation practices, they fall short in three critical aspects when applied to cryptocurrency settings:

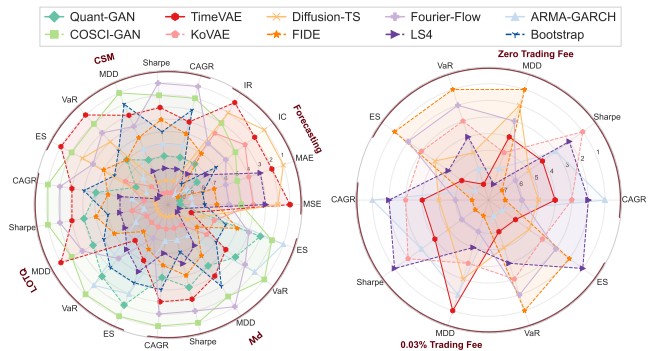

Figure 1: Aggregate rankings of eight TSG models on both tasks from 2021 to 2024: Predictive Utility (left) assesses fidelity and predictive signal quality, and Statistical Arbitrage (right) evaluates trading performance under realistic fee conditions. The results reveal distinct trade-offs across fidelity, tradability, and robustness, with no model uniformly dominating all measures.

- **Limited Domain Generality**: Existing works (Ang et al., 2023a; Hu et al., 2025) focus primarily on traditional assets such as equities and indices (e.g., SPX and CSI300) with lower volatility and restricted trading hours, offering minimal support for cryptocurrency data. They overlook the high-frequency, 24/7 dynamics of crypto markets.
- **Narrow Task Scope**: Most financial time series benchmarks emphasize classification and forecasting, neglecting generation and trading-centric tasks like arbitrage, which are vital for crypto-specific applications such as arbitrage and market-neutral strategies. Moreover, TSG methods in crypto contexts remain largely unexplored.
- **Lack of Crypto-Specific Evaluation**: Existing benchmarks underrepresent measures needed to assess real trading utility. While TSGBench focuses on statistical fidelity, and FinTSB introduces limited financial metrics, both rely on assumptions from traditional markets, failing to consider continuous trading, heavy-tailed risk, and actionable signal quality unique to crypto assets.

To address these limitations, we introduce **CTBench**, the first **C**ryptocurrency **T**ime series generation **Bench**mark. It is an open-source benchmark for rigorous evaluation of TSG methods in cryptocurrency markets, with three key contributions:

- **Crypto-Centric Dataset.** We provide a curated cryptocurrency dataset from major global exchanges, processed via a standardized pipeline with crypto-specific feature support. This ensures analysis-ready data reflecting the volatility and structural nuances of crypto markets.
- **Dual-Task Benchmarks.** To operationalize the utility of synthetic data in real-world finance, CTBench introduces a dual-task evaluation framework that jointly assesses predictive fidelity and tradability. The Predictive Utility task trains forecasters on synthetic data and tests them on real returns, while the Statistical Arbitrage task evaluates whether reconstructed residuals yield tradable mean-reverting signals.
- **Financial Metric Suite.** CTBench introduces a comprehensive evaluation suite over diverse trading strategies spanning forecasting accuracy, rank-based measures, trading performance, and risk metrics, designed to reflect crypto-specific market realities.

We benchmark eight state-of-the-art TSG models and analyze trade-offs across fidelity, tradability, and robustness. Figure 1 shows aggregate rankings across two tasks, with measures radially arranged and averaged over strategies and fee settings. No model dominates universally, highlighting distinct trade-offs between fidelity, tradability, and robustness, and underscoring CTBench's value for informed model selection in crypto trading contexts.

## 2  PRELIMINARIES

Let $\boldsymbol{R} \in \mathbb{R}^{n \times l}$ denote the log-return matrix, where $n$ is the number of tradable crypto-assets and $l$ is the number of hourly return observations. At each time $t \geq 1$, the log-return vector across all assets is $\boldsymbol{r}_t = [r_{1,t}, \cdots, r_{n,t}] \in \mathbb{R}^n$, where each element is defined as $r_{i,t} = \log \frac{p_{i,t}}{p_{i,t-1}}$, with

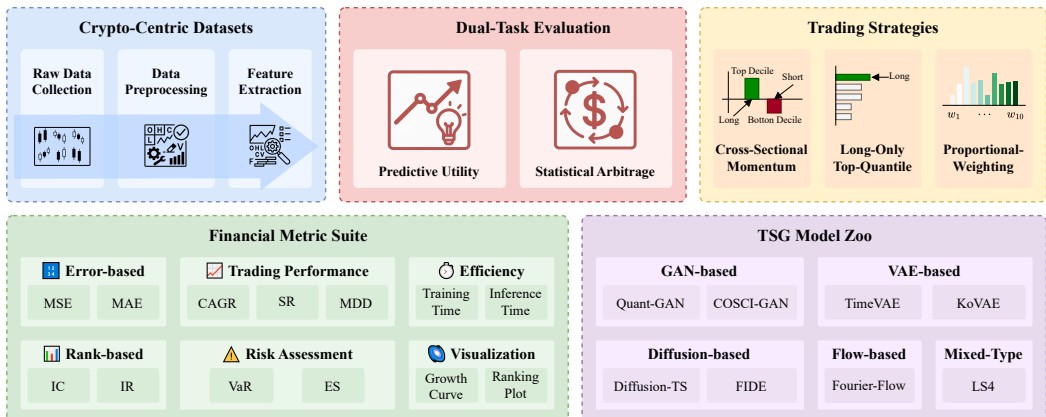

Figure 2: Overall architecture of **CTBench**. The framework unifies five modules: (1) crypto-centric datasets, (2) dual-task evaluation (predictive utility and statistical arbitrage), (3) trading strategies, (4) comprehensive financial metrics, and (5) diverse TSG models into a unified benchmark pipeline.

$p_{i,t}$ the price of asset $i$ at hour $t$. To mimic real-world backtesting, we employ a rolling-window protocol: Given a training window size $w$ and a test step $s$, we define split offsets $\tau \in \mathcal{O} = \{w, w+s, \cdots, w+(k-1)s\}$, where $k = \lfloor \frac{l-w}{s} \rfloor$. Each offset $\tau$ yields a training and test split:

$$\boldsymbol{R}_{\text{train}}^{(\tau)} = [\boldsymbol{r}_{\tau-w+1}, \cdots, \boldsymbol{r}_{\tau}], \ \boldsymbol{R}_{\text{test}}^{(\tau)} = [\boldsymbol{r}_{\tau+1}, \cdots, \boldsymbol{r}_{\tau+s}].$$

For each split, a TSG model $\boldsymbol{g}^{(\tau)}$ is trained on $\boldsymbol{R}_{\text{train}}^{(\tau)}$ and evaluated in two modes: (1) **Generation Mode**: sampling synthetic sequences from Gaussian noise, $\boldsymbol{R}_{\text{gen}} = \boldsymbol{g}^{(\tau)}(\boldsymbol{z})$, $\boldsymbol{z} \sim \mathcal{N}(\boldsymbol{0}, \boldsymbol{I})$; (2) **Reconstruction Mode**: reconstructing training and test set, $\hat{\boldsymbol{R}}_{\text{train}} = \boldsymbol{g}^{(\tau)}(\boldsymbol{R}_{\text{train}}^{(\tau)})$, $\hat{\boldsymbol{R}}_{\text{test}} = \boldsymbol{g}^{(\tau)}(\boldsymbol{R}_{\text{test}}^{(\tau)})$.

We further define a basic portfolio simulation setup: Starting with initial capital $V_0 > 0$, the strategy allocates weights $\boldsymbol{\eta}_t = [\eta_{1,t}, \cdots, \eta_{n,t}] \in \mathbb{R}^n$ at each hour $t$, where $\eta_{i,t}$ denotes the fraction invested in asset $i$. The portfolio value then evolves as $V_t = V_{t-1} \times (\boldsymbol{\eta}_t \cdot \boldsymbol{r}_t)$, with hourly profit-and-loss given by $\Delta V_t = V_t - V_{t-1}$. A summary of notations is provided in Appendix A. To maintain clarity and scope, CTBench restricts its benchmark design to datasets, trading strategies, evaluation measures, and TSG models, as detailed in Appendix B.

## 3 CTBENCH

We present **CTBench**, the first benchmark specifically designed to evaluate Time Series Generation (TSG) models in cryptocurrency markets (Figure 2).

### 3.1 CRYPTO-CENTRIC DATASETS

**Data Overview and Preprocessing.** Our benchmark leverages historical hourly data from all USDT-denominated spot pairs on Binance (Binance Exchange, 2025b), spanning January 2020 to December 2024 and capturing diverse market regimes such as bull runs, crashes, and consolidation phases. To ensure quality, we exclude assets with missing records and retain only USDT pairs, yielding 452 unique cryptocurrencies, a robust foundation for TSG evaluation.

Formally, let $n$ denote the number of tradable crypto assets and $(l+1)$ the number of hourly observations. Each asset $i \in \{1, \cdots, n\}$ at time $t \in \{0, \cdots, l\}$ is represented by four standard fields:

$$\boldsymbol{x}_{i,t} = [O_{i,t}, H_{i,t}, L_{i,t}, C_{i,t}] \in \mathbb{R}^4,$$

where $O$, $H$, $L$, and $C$ are the **Open**, **High**, **Low**, and **Close** prices (quoted in USDT), respectively. Stacking across all assets yields the multi-asset OHLC data array: $\boldsymbol{D} = [\boldsymbol{x}_{i,t}] \in \mathbb{R}^{n \times (l+1) \times 4}$. We focus primarily on **Close** prices and define hourly log-returns as: $r_{i,t} = \log \frac{C_{i,t}}{C_{i,t-1}}$, where $t \in \{1, \cdots, l\}$, giving the return matrix $\boldsymbol{R} \in \mathbb{R}^{n \times l}$.

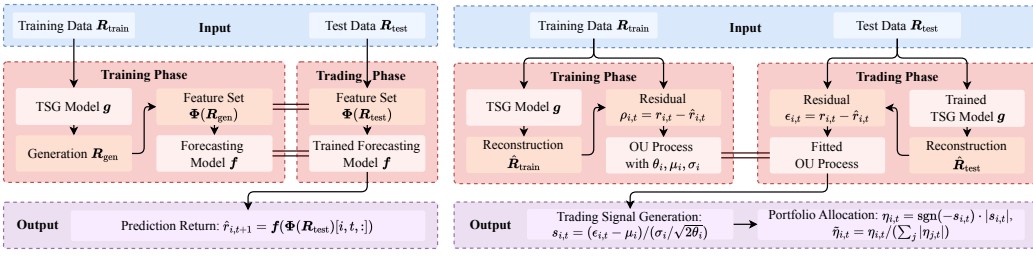

(a) Predictive Utility task.          (b) Statistical Arbitrage task.

Figure 3: Architectures of dual-task benchmarks.

**Feature Extraction.** To capture key market dynamics, we extract $d$ features widely used in quantitative trading, such as Alpha101 factors (Kakushadze, 2016), Bollinger Bands, RSI, and moving averages (Sun et al., 2023; Zhu & Zhu, 2025; Zhang et al., 2020; Tsai & Hsiao, 2010). These features encode signals such as momentum, mean-reversion, and volatility. Applying the same pipeline to both real and synthetic data enables consistent evaluation of TSG models' ability to replicate the statistical and structural properties vital for downstream tasks. Formally, let $\boldsymbol{\Phi} = \{\phi_j\}_{j=1}^{d}$ be the feature set, where each $\phi_j : \mathbb{R}^{n \times l} \to \mathbb{R}^{n \times l}$ acts on the return matrix $\boldsymbol{R}$. Applying $\boldsymbol{\Phi}$ yields a feature tensor with shape $\mathbb{R}^{n \times l \times d}$. The dataset exhibits strong cross-sectional dispersion and cap-dependent volatility, motivating crypto-specific evaluation of predictive structure and cross-asset dynamics beyond marginal similarity (see Appendix C.1).

## 3.2 DUAL-TASK EVALUATION

To connect generation fidelity with financial utility, CTBench introduces dual-task evaluation assessing both predictive realism and tradable structure. As shown in Figure 3, these tasks measure whether synthetic data preserves useful forecasting signals (predictive utility) or enables the discovery of stationary, market-neutral alpha (statistical arbitrage). Details are in Appendix C.2.

**Predictive Utility Task.** This task tests whether synthetic data can train forecasters that generalize to real markets. As shown in Figure 3(a), given a training log-return window $\boldsymbol{R}_{\text{train}}^{(\tau)}$, a TSG model $\boldsymbol{g}$ generates synthetic returns $\boldsymbol{R}_{\text{gen}} = \boldsymbol{g}(\boldsymbol{z})$, $\boldsymbol{z} \sim \mathcal{N}(\boldsymbol{0}, \boldsymbol{I})$. Features $\boldsymbol{\Phi}(\boldsymbol{R}_{\text{gen}})$ are extracted to train a forecasting model $\boldsymbol{f}$, instantiated as XGBoost (Chen & Guestrin, 2016) for its robustness to heterogeneous and noisy financial features (Vancsura et al., 2025; Liu et al., 2021; Yun et al., 2021). The ablation in §4.4 shows that XGBoost best balances low prediction error and strong cross-sectional ranking fidelity, making it the most informative evaluator for trading utility.

The trained $\boldsymbol{f}$ is deployed on real test data $\boldsymbol{R}_{\text{test}}$ to generate signals for a dollar-neutral long-short portfolio, rebalanced hourly over a month. This setup tests whether $\boldsymbol{R}_{\text{gen}}$ preserves predictive signals with measurable economic value. All components, the TSG model, features, and the forecaster, are modular, allowing extensibility across architectures.

**Statistical Arbitrage Task.** In contrast to the generation-focused task, this task evaluates whether TSG models can reconstruct market structure and isolate tradable, mean-reverting residuals for statistical arbitrage. As depicted in Figure 3(b), a model $\boldsymbol{g}$ is trained on real returns $\boldsymbol{R}_{\text{train}}$ to produce reconstructions $\hat{\boldsymbol{R}}_{\text{train}}$, and the residuals $\rho_{i,t} = r_{i,t} - \hat{r}_{i,t}$ are assumed to follow Ornstein–Uhlenbeck (OU) processes (Uhlenbeck & Ornstein, 1930), parameterized by estimated $(\mu_i, \theta_i, \sigma_i)$ per asset.

On test data, new residuals $\epsilon_{i,t}$ are mapped to standardized $s$-scores $s_{i,t} = (\epsilon_{i,t} - \mu_i)/(\sigma_i/\sqrt{2\theta_i})$, which drive trading decisions via thresholding ($\gamma = 2$) and weight normalization. Portfolios are rebalanced hourly based on these signals. This task complements generation-focused evaluation by assessing the model's ability to reveal stationary, market-neutral alpha, thus bridging statistical fidelity and practical trading utility.

## 3.3 TRADING STRATEGIES

CTBench is strategy-agnostic, evaluating TSG models across diverse trading paradigms. Profitability and risk metrics (§3.4) are computed uniformly for all backtests, enabling rigorous stress testing

whether models capture genuine market structure rather than overfitting specific strategies. We evaluate models under three canonical trading strategies commonly adopted in cryptocurrency markets:

- **Cross-Sectional Momentum (CSM):** take long positions in the top decile of predicted assets while shorting the bottom decile, capturing relative momentum effects.
- **Long-Only Top-Quantile (LOTQ):** build an equal-weight portfolio of the top 20% of assets, reflecting long-biased strategies often favored in practice.
- **Proportional Weighting (PW):** allocate capital in proportion to predicted returns, directly translating forecasts into position sizes.

This modular design supports plug-and-play integration of additional or proprietary strategies. Full details are provided in Appendix C.3.

## 3.4 FINANCIAL METRIC SUITE

Evaluating TSG models for financial applications requires more than mere statistical similarity, it demands measuring whether synthetic data enables profitable and risk-aware trading. To this end, CTBench defines thirteen core metrics, grouped into six practitioner-relevant categories:

- **Error-based Evaluation:** At the most fundamental level, do synthetic returns numerically resemble real ones? Metrics include Mean Squared Error (**MSE**), which emphasizes volatility mismatches, and Mean Absolute Error (**MAE**), which is more robust to outliers.
- **Rank-based Evaluation:** Do synthetic returns preserve relative asset ordering? Information Coefficient (**IC**) measures rank correlation, and Information Ratio (**IR**) evaluates its stability.
- **Trading Performance:** Does synthetic data yield actionable, profitable signals? Statistical accuracy does not guarantee financial profitability. We therefore use Compound Annual Growth Rate (**CAGR**) for long-term growth and Sharpe Ratio (**SR**) for return-to-risk balance.
- **Risk Assessment Metrics:** Do models capture fat tails and downside risks? We compute Maximum Drawdown (**MDD**) for worst-case loss, Value at Risk (**VaR**) at 95% confidence, and Expected Shortfall (**ES**) for tail risk beyond VaR.
- **Efficiency:** Can models support real-time deployment? We track **Training Time** and **Inference Time** to assess adaptability in fast-moving crypto markets.
- **Visualization:** Do results exhibit contextual realism? We report **Simulated Growth Curves** for a $10,000 investment and cross-sectional **Ranking Plots** across market regimes to illustrate interpretability and contextual realism.

Together, these metrics ensure balanced evaluation of fidelity, utility, and practicality. Full definitions are in Appendix C.4.

## 3.5 TSG MODEL ZOO

Generative models for time series aim to capture temporal dependencies and structural patterns, with backbones spanning GANs, VAEs, diffusion models, flow models, and mixed-type models (Ang et al., 2023a; Nikitin et al., 2023) (see Table 3). Yet, nearly half of prior TSG works do not evaluate in financial contexts, and those that do typically focus on traditional markets such as equities or macroeconomic data, leaving a gap for cryptocurrency applications. To close this gap, CTBench evaluates eight state-of-the-art models spanning five families:

- **GANs: Quant-GAN** (Wiese et al., 2020) and **COSCI-GAN** (Seyfi et al., 2022), applied only in forecasting tasks since GANs do not natively support reconstruction (Goodfellow et al., 2020).
- **VAEs: TimeVAE** (Desai et al., 2021) and **KoVAE** (Naiman et al., 2024b), which extend variational autoencoders for temporal dynamics.
- **Diffusion Models: Diffusion-TS** (Yuan & Qiao, 2024) and **FIDE** (Galib et al., 2024), leveraging denoising-based generative processes for time series.
- **Flow-based Models: Fourier-Flow** (Alaa et al., 2021)), employing invertible transformations for likelihood-based generation.
- **Mixed-type Models: LS4** (Zhou et al., 2023), designed to unify strengths across multiple generative paradigms.

These models are selected for their generative fidelity and practical relevance to financial downstream tasks. Further details are in Appendix C.5.

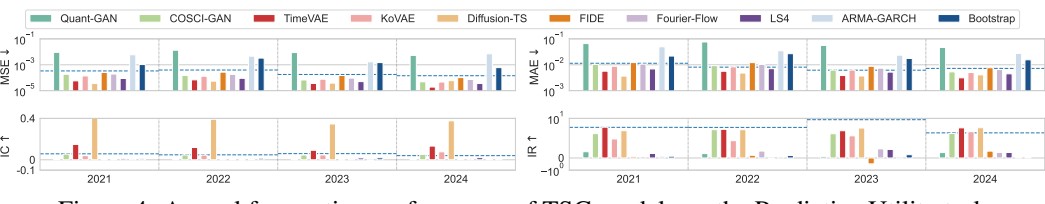

Figure 4: Annual forecasting performance of TSG models on the Predictive Utility task.

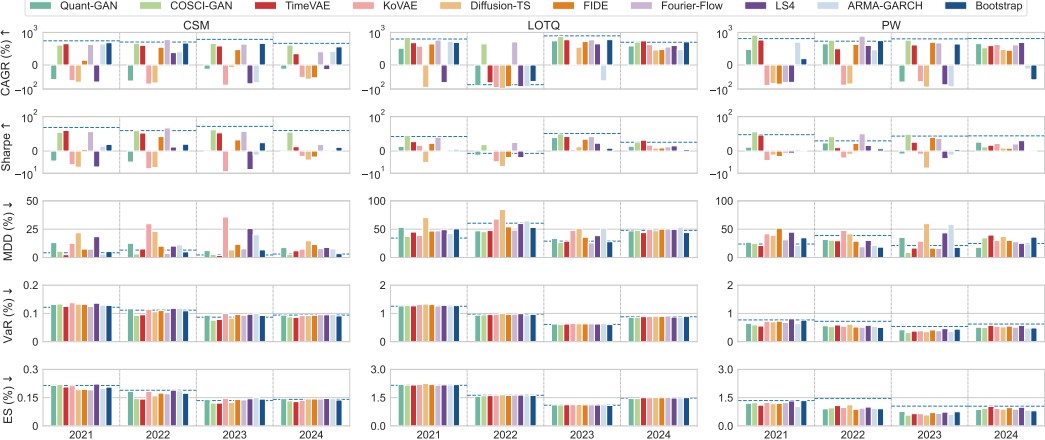

Figure 5: Annual trading performance of TSG models on the Predictive Utility task.

## 4 EXPERIMENTS

We evaluate CTBench on 452 USDT trading pairs using a walk-forward setup: each cycle trains on 500 days and tests on 30 days (Predictive Utility) or 15 days (Statistical Arbitrage), with retraining before every window. To isolate generator quality, transaction fees are zero by default; for Statistical Arbitrage, we also report results under a 0.03% fee reflecting typical exchange costs Zhang et al. (2023); Winkel & Härdle (2023); Binance Exchange (2025a). We benchmark eight TSG models across five architectural families and include two classical baselines widely used in quantitative finance: **ARMA-GARCH** (Engle, 1982), evaluated on both tasks, and a **Bootstrap** generator (Rubin, 1981), used only for Predictive Utility. All models follow recommended or stable hyperparameters and are scored using CTBench's full financial metric suite (details in Appendix D).

### 4.1 PREDICTIVE UTILITY TASK

Figures 4 and 5 report year-wise forecasting and trading performance from 2021 to 2024. The blue dashed line denotes the baseline using real data (without TSG), whose consistently strong returns validate the effectiveness of our feature-extraction pipeline (**§3.1**).

**Annual Predictive Utility Analysis.** Across all four market regimes, predictive accuracy and financial profitability often diverge, underscoring the difficulty of converting statistical fidelity into tradable signals. In **2021 (Bull Market)**, Diffusion-TS achieves the best forecasting error yet fails to convert it into returns, while TimeVAE and COSCI-GAN demonstrate strong risk-adjusted performance by balancing denoising and alpha amplification. In **2022 (High Volatility)**, TimeVAE remains resilient, and COSCI-GAN benefits from dispersion, whereas Diffusion-TS struggles with frequent reversals. During **2023 (Consolidation)**, Fourier-Flow outperforms in risk-adjusted metrics due to its frequency-preserving structure, while trend-reliant models degrade. By **2024 (Low-Signal / Mean Reversion)**, Most models face diminishing signal strength; only TimeVAE retains marginal profitability. Classical baselines ARMA-GARCH and Bootstrap maintain mid-tier performance: solid risk control but limited forecasting capability. Overall, results show that synthetic data must align with market structure and strategy design, not just minimize reconstruction error.

**Ranking Analysis.** Radar-plot rankings in Figure 6 reveal three consistent patterns: (1) **Diffusion-TS** achieves the strongest forecasting scores but performs poorly in trading, illustrating a common gap between statistical fidelity and economic usefulness. (2) **TimeVAE** and **COSCI-GAN** show

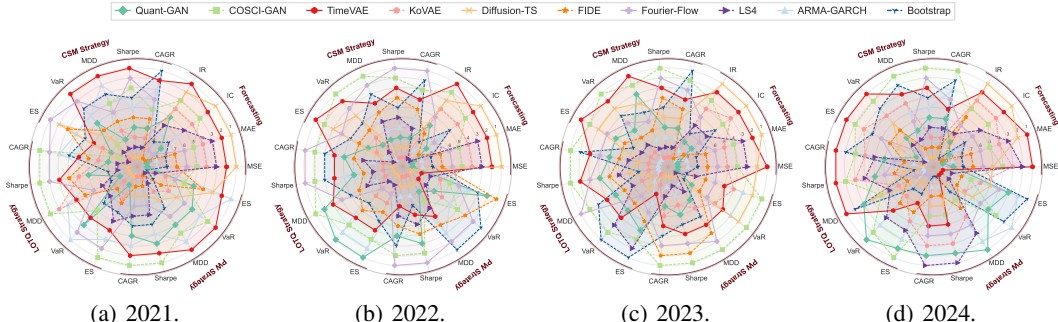

(a) 2021.     (b) 2022.     (c) 2023.     (d) 2024.

Figure 6: Annual rankings of TSG models on the Predictive Utility task.

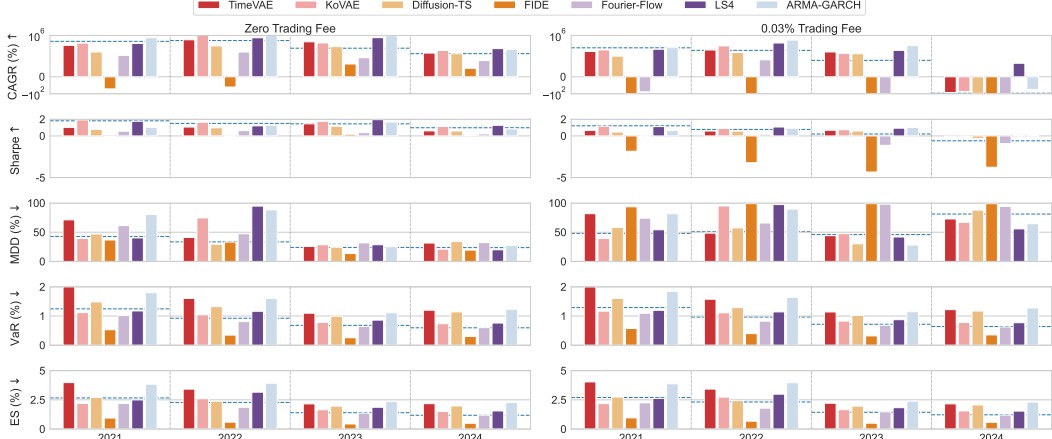

Figure 7: Annual performance of TSG models on the Statistical Arbitrage task.

regime-dependent strengths: TimeVAE excels in stable or mean-reverting markets due to its regularization, while COSCI-GAN benefits from volatile, directional regimes where amplified variance enhances trend signals. (3) **Fourier-Flow** maintains steady mid-to-high rankings across categories, positioning it as a robust, all-weather choice for risk-managed deployment.

These trends reinforce a core insight: *low reconstruction or prediction error does not ensure trading success*. Over-regularization can suppress alpha-bearing variance, while models that preserve structural noise (TimeVAE, COSCI-GAN) produce more actionable signals. Effective model choice must consider market regime and strategy alignment.

**Equity Curve Dynamics.** Figure 14 presents log-scaled equity curves from 2021–2024 for each TSG model across three trading strategies, highlighting how model inductive biases shape cumulative returns. COSCI-GAN consistently captures directional gains, while TimeVAE and Fourier-Flow provide smoother but moderate returns. Diffusion-based models underperform due to volatility suppression, and LS4 remains conservatively flat. The ARMA-GARCH baseline similarly yields largely flat. The Bootstrap generator typically sits in the upper middle of the pack, with equity curves that track the top 3-5 models. Full results and visualizations are in Appendix E.1.

## 4.2 STATISTICAL ARBITRAGE TASK

Figure 7 reports annualized trading and risk metrics under both zero-fee and realistic-fee conditions. The blue dashed line represents a Principal Component Analysis (PCA) baseline trained on $R_{\text{train}}$, reflecting a standard statistical-arbitrage baseline used as a reference for TSG evaluation.

**Annual Performance Analysis.** All models exhibit reduced profitability under transaction costs, with the impact tied to turnover. Among TSG models, **KoVAE** excels in volatile regimes with high but mean-reverting swings, whereas **LS4** achieves strong returns with stable drawdowns, particularly in 2023. Even after fees, both maintain leading positions, underscoring the value of regime adaptability and cost-aware design. **TimeVAE** and **Diffusion-TS** form a second tier, trading off peak returns for smoother, fee-resilient SR, though their exposure to tail risk, especially in 2021 and

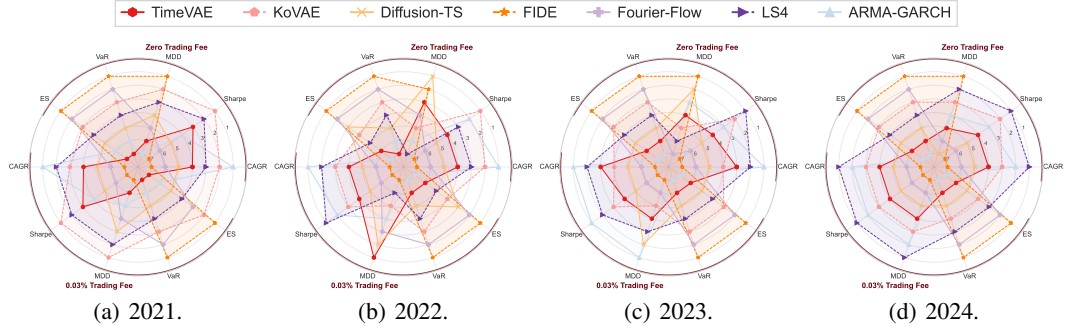

Figure 8: Annual rankings of TSG models on the Statistical Arbitrage task.

2024, limits overall efficiency. **FIDE** consistently yields negligible or negative returns but achieves the lowest VaR, ES, and MDD, suggesting over-regularized residuals that suppress tradable variance. **Fourier-Flow** similarly underperforms, failing to capture mean-reverting structure despite effective noise smoothing, highlighting the limitations of exact-likelihood flow models in arbitrage-centric tasks. **ARMA-GARCH** delivers the top CAGR across 2021 to 2023, but consistently exhibits the weakest tail-risk profile, with the worst VaR and ES among all models.

**Ranking Analysis.** Figure 8 visualizes annual model rankings via radar plots, revealing how TSG models balance return, risk, and stability across regimes. **KoVAE** and **LS4** form polygons that strongly protrude along CAGR and Sharpe but collapse along risk dimensions, indicating high returns paired with elevated drawdown and tail exposure, especially in turbulent years. **FIDE** shows the opposite pattern: tight risk control but consistently weak returns, reflecting a capital-preserving yet alpha-limited design. **TimeVAE** and **Diffusion-TS** produce more rounded, balanced shapes with neither dominant peaks nor severe failures, suggesting steady, regime-agnostic robustness.

Introducing trading fees compresses the rank distances: high-turnover models (e.g., KoVAE) lose several Sharpe positions, whereas smoother, lower-turnover models (e.g., TimeVAE, Diffusion-TS) retain most of their ranking. This highlights an important practical insight: models that generate smoother residual signals naturally incur lower costs and therefore achieve better fee-adjusted performance. Finally, the year-to-year evolution of polygon shapes reveals regime sensitivity: LS4 expands sharply on CAGR in 2023 but suffers high drawdowns in 2022, while KoVAE excels in volatile periods yet underperforms in calmer markets. These dynamics emphasize that model selection must consider both regime characteristics and operational constraints.

**Equity Curve Dynamics.** Figure 15 illustrates fee-adjusted equity curves starting at $10,000 with 0.03% trading fees. LS4 and KoVAE deliver sustained growth, while TimeVAE plateaus in later regimes. Diffusion-TS is stable but low-return; FIDE and Fourier-Flow

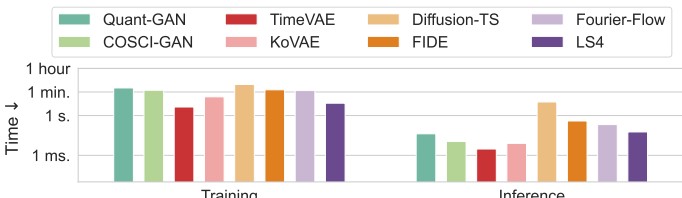

Figure 9: Training and inference time of TSG models.

underperform due to overly smoothed residuals. Remarkably, ARMA-GARCH exhibits moderate gains through early 2023 but experiences substantial growth afterward, ultimately becoming the top-performing model by 2024-2025. Full results and visualizations are detailed in Appendix E.2.

## 4.3 EFFICIENCY

We compare training and inference times for all TSG models in Figure 9. VAE-based models are the most efficient: **TimeVAE** trains in under a minute and achieves sub-second inference, making it ideal for real-time applications, low-latency backtesting, and rapid retraining in fast-moving markets. GAN-based models offer moderate efficiency: **COSCI-GAN** maintains balanced runtime, while **Quant-GAN** is significantly more expensive to train with no corresponding speed advantage at inference. Diffusion-based models are the slowest: **Diffusion-TS** incurs the longest runtime due to iterative denoising, and **FIDE** provides only modest improvements. Despite their strong fidelity

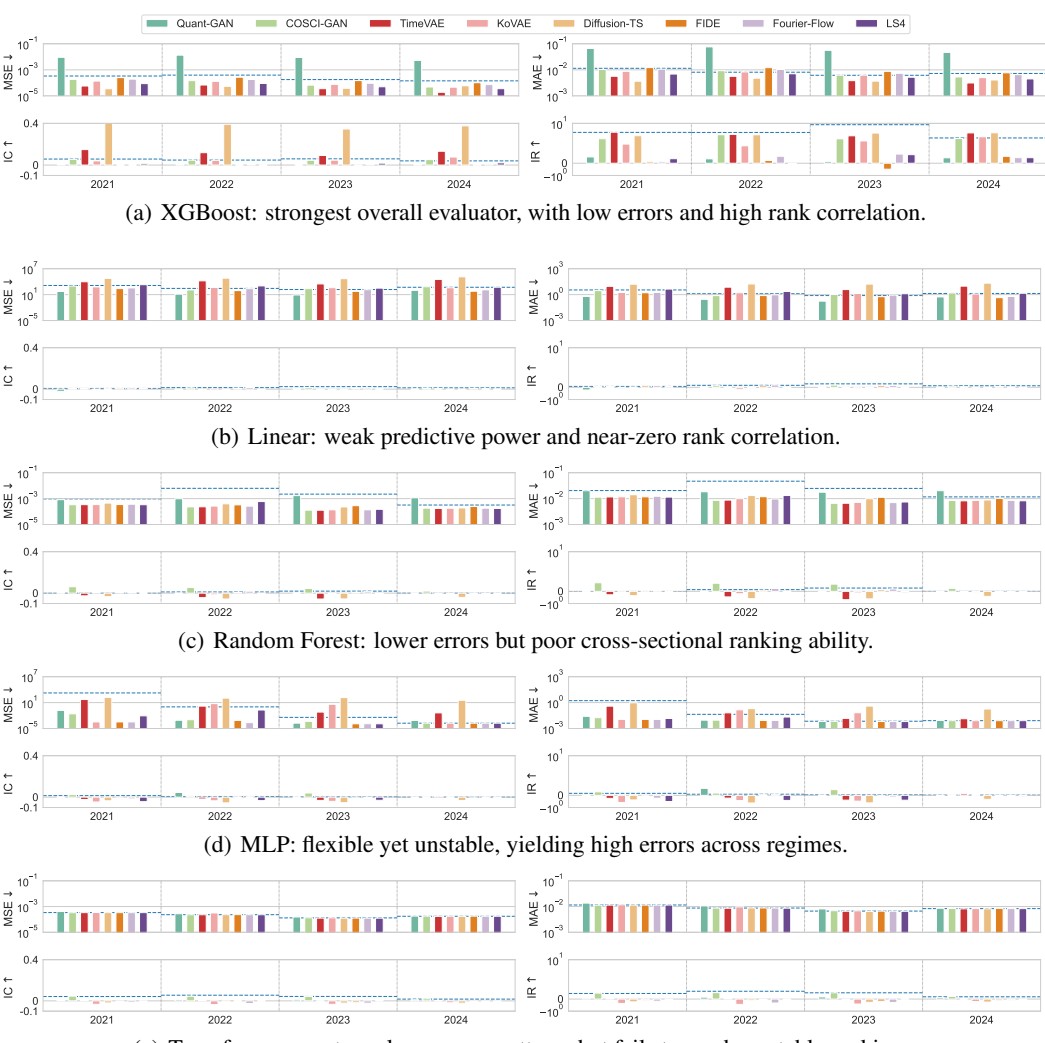

(a) XGBoost: strongest overall evaluator, with low errors and high rank correlation.

(b) Linear: weak predictive power and near-zero rank correlation.

(c) Random Forest: lower errors but poor cross-sectional ranking ability.

(d) MLP: flexible yet unstable, yielding high errors across regimes.

(e) Transformer: captures long-range patterns but fails to produce stable rankings.

Figure 10: Ablation on the choice of forecasting model in the Predictive Utility task. We compare five forecasters: XGBoost, Linear Regression, Random Forest, MLP, and Transformer, to assess whether TSG-model rankings remain stable across evaluators.

and risk–return profiles, these models are best suited for offline or compute-rich settings. Flow-based and mixed-type models occupy the middle ground, offering reasonable efficiency and reliable likelihood calibration, though their latency limits real-time use. Overall, **VAE-based models** and **LS4** emerge as the most practical choices for latency-sensitive or resource-constrained deployment, whereas diffusion-based models are better reserved for offline or batch-generation pipelines.

## 4.4 ABLATION STUDY ON FORECASTING MODELS

A crucial component of the Predictive Utility task is the choice of forecasting model used to assess the quality of synthetic data. In CTBench, we adopt XGBoost (Chen & Guestrin, 2016) as the default forecaster due to its strong empirical performance in quantitative finance and its robustness to heterogeneous, noisy features, properties particularly well aligned with crypto markets (Vancsura et al., 2025; Liu et al., 2021; Yun et al., 2021).

To validate this design choice, we conduct an ablation study comparing five forecasting architectures: XGBoost, Linear Regression (Hilt & Seegrist, 1977), Random Forest (Breiman, 2001), a Multi-Layer Perceptron (MLP) (Hinton, 1990), and a Transformer-based sequence model (Vaswani et al., 2017). Each forecaster is trained on features extracted from identical sets of synthetic data, ensuring that performance differences arise solely from the forecasting backbone rather than the

Table 1: Scenario-based recommendations for selecting TSG models in cryptocurrency markets.

| Scenario | Recommendation | Rationale |
|---|---|---|
| Trend-following / Directional Markets | COSCI-GAN, KoVAE | COSCI-GAN amplifies trend and dispersion; KoVAE offers alpha with higher drawdowns |
| Mean-reverting / Range-bound Regimes | TimeVAE, Fourier-Flow, Diffusion-TS | TimeVAE / Fourier-Flow provide balance; Diffusion-TS preserves rank order |
| Fee-sensitive / Low-turnover Settings | TimeVAE, Diffusion-TS | Smooth residuals, stable Sharpe under transaction costs |
| Risk Tolerance / Portfolio Design | KoVAE, LS4, TimeVAE, Diffusion-TS, FIDE | KoVAE / LS4 maximize returns with risk; TimeVAE / Diffusion-TS balance Sharpe and drawdown; FIDE is defensive |
| Deployment Efficiency | TimeVAE, LS4 | Fast retraining and low-latency inference; diffusion models better suited for offline use |

underlying TSG model. Evaluation follows the same walk-forward protocol as the main benchmark and uses both error-based metrics (MSE, MAE) and rank-based metrics (IC, IR).

Figure 10 summarizes the results. **Linear Regression** and **MLP** consistently exhibit high MSE and MAE, indicating poor point-wise prediction accuracy. **Random Forest** and **Transformer** reduce point-wise errors but fail to preserve cross-sectional ordering, both achieving near-zero IC/IR, suggesting that they primarily fit short-term fluctuations rather than tradable structure. In contrast, **XGBoost** achieves the strongest balance: low prediction error paired with robust rank correlations, making it significantly more sensitive to the differences across TSG models and more aligned with how trading strategies consume forecasts.

Overall, the ablation demonstrates that **XGBoost offers the most discriminative and trading-relevant evaluation signal**, supporting its use as the default forecaster in CTBench. It consistently reveals meaningful differences between TSG models while avoiding the instability or rank collapse seen in alternative forecasters.

### 4.5 RECOMMENDATIONS

Our findings reveal a four-way trade-off among TSG model families: (1) VAE-based models ensure stable reconstruction but might under-react to fast-changing regimes. (2) GAN-based approaches extract trend alpha but suffer from volatility-induced instability. (3) Diffusion models handle regime clustering and fat tails well, but degrade under low signal regimes. (4) Flow-based models prioritize likelihood but offer limited utility, while mixed-type ones are efficient but inconsistent in risk-return.

Based on these findings, Table 1 presents actionable guidelines for practitioners. These recommendations help align model choice with market conditions, strategic objectives, and operational constraints. Crucially, there is **no one-size-fits-all TSG model** for cryptocurrency markets. Instead, practitioners should: (1) diagnose the prevailing market regime, alpha source, and system constraints; (2) select a model whose inductive bias strengthens the target structure while preserving tradability; and (3) evaluate performance using task–metric pairs that best reflect production goals. By integrating dual-task design with a comprehensive evaluation suite, CTBench provides the decision surface needed to navigate these choices effectively.

## 5 CONCLUSION AND FUTURE WORK

We present CTBench, the first benchmark explicitly designed for time series generation in cryptocurrency markets, combining a high-frequency token dataset, a dual-task evaluation framework, and a comprehensive suite of financial metrics to assess both statistical fidelity and practical viability. Empirical results reveal key trade-offs across TSG families and highlight strategy-dependent model behaviors, offering actionable insights for real-world deployment.

Looking ahead, we plan to enhance CTBench by incorporating more advanced forecasters in the Predictive Utility task and integrating richer residual modeling processes for the Statistical Arbitrage task. Future directions also include expanding to a broader set of tokens, incorporating exogenous signals such as trading volume, and benchmarking more sophisticated generative architectures.

ACKNOWLEDGMENTS

This research/project is supported by the National Natural Science Foundation of China (NSFC) under Grant No. U25B6003, the National Research Foundation, Singapore, under its AI Singapore Programme (AISG Award No: AISG3-RP-2022-029) and CyberSG R&D Programme (CRPO Award No: CRPO-GC5-NUS-006), the Ministry of Education, Singapore, under its MOE AcRF TIER 1 Grant (T1 251RES2517), and the National University of Singapore, School of Computing Seed Fund.

ETHICS STATEMENT

CTBench is developed to improve the rigor, transparency, and practical relevance of TSG evaluation in financial contexts. By integrating realistic trading tasks, crypto-specific risk metrics, and an open, reproducible benchmarking pipeline, our goal is to promote more responsible and economically grounded research at the intersection of machine learning and quantitative finance.

**Data Usage and Privacy.** All datasets are publicly available cryptocurrency market data at the asset level (e.g., prices and volumes). The dataset contains no personally identifiable information, private user data, or proprietary trading records. The benchmark operates exclusively on aggregate market statistics, ensuring compliance with standard data privacy and ethical research practices.

**Bias, Market Impact, and Responsible Use.** As with any financial system, models trained on historical market data may inherit structural patterns, regime dependencies, or latent biases present in past market behavior. While CTBench evaluates statistical fidelity and economic utility, it does not guarantee robustness under extreme market events or structural breaks. We encourage further research on robustness, stress testing, regime adaptation, and responsible deployment of AI systems in volatile financial environments. Additionally, synthetic data should be used cautiously, particularly in settings where model outputs could influence capital allocation or market dynamics.

**Financial Use Disclaimer.** This work is intended solely for academic research. The models, strategies, and results are presented to evaluate benchmarking methodology rather than to provide investment guidance. Nothing herein constitutes financial advice, investment recommendation, or endorsement of any asset, exchange, or trading strategy. CTBench should not be used for live trading or financial decision-making without independent validation and regulatory compliance.

REPRODUCIBILITY STATEMENT

We are committed to ensuring the transparency and reproducibility of CTBench by providing open resources, detailed documentation, and standardized evaluation protocols.

**Source Code.** To facilitate replication and future extensions, we release the full implementation at: `https://github.com/MilleXi/CTBench/`. The repository contains all core components of CTBench, including data loaders, preprocessing pipelines, task definitions, evaluation metrics, and benchmarking scripts for TSG models. This enables end-to-end reproduction of both tasks and reported results.

**Datasets.** All cryptocurrency datasets used in CTBench are publicly accessible. We provide detailed documentation of token selection criteria, time ranges, filtering rules, and preprocessing procedures in Section 3.1 and Appendix C.1. These descriptions allow researchers to reconstruct the dataset directly from raw public sources.

**Benchmark Protocol and Evaluation.** CTBench evaluates eight representative TSG models under two complementary tasks: Predictive Utility and Statistical Arbitrage. Task formulations, model implementations, and experimental settings are described in Section 4 and Appendices C–D. All evaluation metrics, including error-based, rank-based, trading, risk, and efficiency measures, are formally defined in Section 3.4 and Appendix C.4. Key hyperparameters for each model are reported in Appendix D to ensure fair comparison and reproducibility.

**Summary.** Together, these resources provide a complete and extensible toolkit for reproducing our experiments and supporting future research on time-series generation in financial domains.

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

## A  NOTATIONS

Table 2 summarizes the symbols and definitions used throughout the paper for quick reference.

## B  SCOPE ILLUSTRATION

To ensure clarity and fairness in benchmarking, CTBench explicitly defines its scope across datasets, strategies, evaluation measures, and model families. This scoped design avoids confounding factors, emphasizes crypto-specific characteristics, and provides a standardized basis for comparing diverse TSG methods. The followings detail these design choices.

Table 2: List of frequently used notations.

| Symbol | Description |
|--------|-------------|
| $\boldsymbol{R} \in \mathbb{R}^{n \times l}$ | Log-return matrix with $n$ assets and $l$ hourly observations |
| $\boldsymbol{r}_t = [r_{i,t}] \in \mathbb{R}^n$ | Log-return vector of time $t$ of all $n$ assets |
| $w, s, k, \tau$ | Training window size, test step, # of splits, split offset |
| $\boldsymbol{R}_{\text{train}}, \boldsymbol{R}_{\text{test}}$ | Training data of returns, test data of returns |
| $\boldsymbol{g}$ | Time series generation (TSG) model |
| $\boldsymbol{R}_{\text{gen}}, \hat{\boldsymbol{R}}_{\text{train}}, \hat{\boldsymbol{R}}_{\text{test}}$ | Generated time series, reconstruction of train and test sets |
| $\boldsymbol{\eta}_t = [\eta_{i,t}] \in \mathbb{R}^n$ | Portfolio weight vector at hour $t$ |
| $V_0, V_t, \text{and } \Delta V_t$ | Initial capital, portfolio equity, and profit-and-loss at hour $t$ |
| $O, H, L, C$ | Open, High, Low, and Close (OHLC) prices |
| $\boldsymbol{D} = [\boldsymbol{x}_{i,t}]$ | Multi-asset OHLC data array |
| $\boldsymbol{\Phi} = \{\boldsymbol{\phi}_j\}_{j=1}^d$ | A feature set $\boldsymbol{\Phi}$ with $d$ feature mapping function $\boldsymbol{\phi}_j$ |

**Scope of Datasets.** CTBench is restricted to cryptocurrency markets due to their unique properties, such as 24/7 trading, high volatility, and fragmented liquidity. In selecting the concrete dataset, we focus on Binance USDT spot pairs (Binance Exchange, 2025b) because Binance is consistently ranked as the largest centralized exchange by global spot volume and provides broad, liquid coverage of actively traded assets. Its freely accessible, high-quality historical data ensures that CTBench remains fully reproducible without requiring proprietary data contracts.

Moreover, we use only raw time series inputs (i.e., returns), excluding side-channel information (e.g., order books, blockchain logs, or news). This isolates core generative capabilities without reliance on auxiliary signals. We employ only well-established financial features (e.g., Alpha101 (Kakushadze, 2016)) to ensure compatibility with real-world quantitative trading while minimizing noise from complex feature engineering.

**Scope of Trading Strategies.** To capture diverse trading behaviors, we benchmark TSG models across three canonical strategies, ranging from rank-based to magnitude-sensitive and from directional to market-neutral setups. This ensures a holistic evaluation of whether synthetic data generalizes across real-world trading paradigms or merely overfits to specific signal patterns.

**Scope of Evaluation Measures.** Our benchmark incorporates a curated set of evaluation measures widely recognized in financial TSG research (Ang et al., 2023a; Wiese et al., 2020), ensuring a holistic assessment of statistical fidelity and financial utility. We have excluded metrics with limited practical relevance or interpretability to maintain a focused and meaningful evaluation framework for the crypto domain.

**Scope of TSG Models.** We select TSG models capable of handling multivariate inputs typical in crypto markets, encompassing both general-purpose and finance-specific architectures. Our selection spans five model families: GAN, VAE, diffusion, flow, and mixed-type, favoring architectures with general applicability over domain-specific requirements. All models are trained under a unified protocol without excessive hyperparameter tuning to ensure fair benchmarking and reflect practical deployment constraints.

## C  BENCHMARK DETAILS

### C.1  CRYPTO-CENTRIC DATASETS

**Data Descriptive Statistics.** Understanding the statistical profile of crypto returns is essential for designing effective TSG benchmarks. We analyze the distribution of log-returns to identify deviations from normality, such as skewness and kurtosis, stylized facts well documented in financial time series. Cryptocurrencies, in particular, often exhibit **fat-tailed distributions**, indicating elevated probability of extreme price movements.

Figure 11 presents histograms of the mean hourly log-return and mean hourly volatility (standard deviation of log-returns) across all 452 cryptocurrencies. The mean hourly returns are centered around zero but show a slight right skew, suggesting modestly positive drift in most assets. In

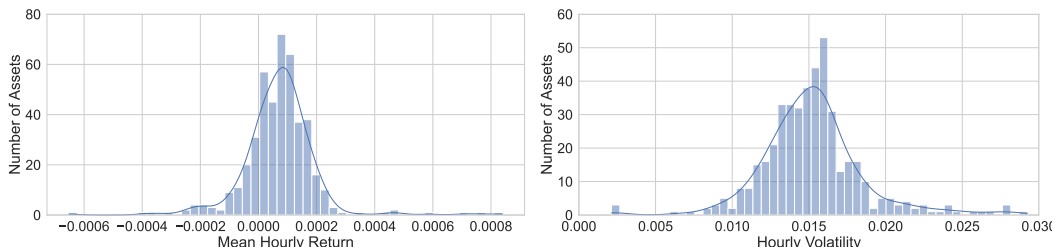

Figure 11: Histograms of the mean hourly log-return (%) (left) and mean hourly volatility (%) (right).

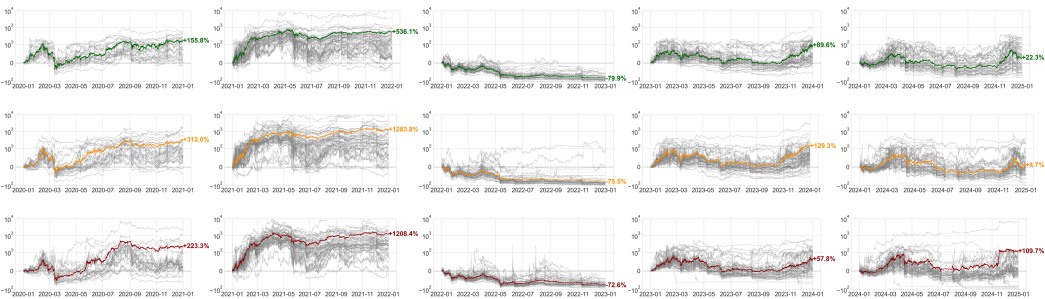

Figure 12: Line plots of closing returns for representative cryptocurrencies, with large-cap examples (top row), mid-cap examples (middle row), and small-cap examples (bottom row), displayed annually from 2020 to 2024.

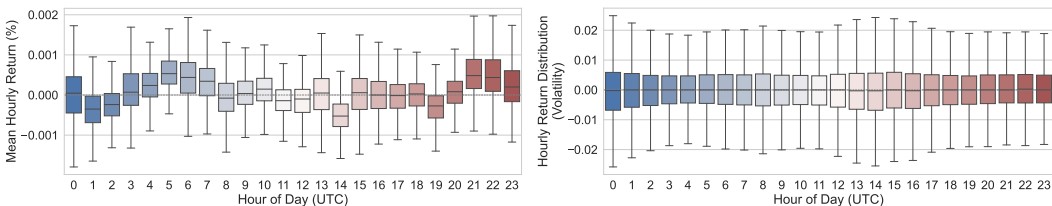

Figure 13: The mean hourly log-return (%) (left) and mean hourly volatility (%) (right) by hour of day (UTC).

contrast, the mean hourly volatility exhibits a long right tail, indicating that while many assets trade with low volatility, a notable subset experiences highly volatile price swings.

To visualize market dynamics over time, we categorize cryptocurrencies into large-, mid-, and small-cap groups and plot representative closing prices annually from 2020 to 2024 in Figure 12. These trajectories highlight significant market regimes, including the bull runs of 2020–2021, sharp corrections in 2022, and subsequent periods of recovery or consolidation. Notably, mid- and small-cap assets often display greater volatility and sharper price swings than their large-cap counterparts.

Given that cryptocurrency markets operate 24/7, intraday patterns provide valuable insights into market microstructure. Figure 13 depicts the mean hourly log-return and volatility by time of day. We observe return peaks around early morning (5–7 AM) and late evening (9–11 PM), reflecting heightened trading during transitions between major global financial centers. Volatility peaks notably around midnight and during overlapping trading hours between US and Europe (12–5 PM), suggesting periods of intensified market activity driven by global participation and algorithmic strategies.

**Discussions.** Our analysis reveals several critical insights shaping the design of CTBench:

- **Complex Market Dynamics:** Crypto markets exhibit high-frequency, high-dimensional behaviors with distinct volatility profiles, intraday cycles, and regime shifts. These factors necessitate benchmarks tailored for crypto time series.
- **Benchmark Task Design:** Given the data's complexity, evaluation tasks must probe whether synthetic data preserves predictive structures critical for practical applications such as forecasting and statistical arbitrage.

- **TSG Model Requirements:** Capturing the intricate temporal and cross-sectional dependencies of crypto markets demands advanced TSG architectures capable of modeling both short-term fluctuations and long-term trends.
- **Evaluation Metrics:** Assessing TSG performance in crypto markets requires multifaceted metrics that go beyond statistical fidelity to capture financial viability and risk sensitivity.

Collectively, these insights underscore the need for crypto-specific benchmarks like CTBench to advance the evaluation and development of TSG models for this rapidly evolving domain.

## C.2 DUAL-TASK EVALUATION

### C.2.1 PREDICTIVE UTILITY TASK

**Training Phase.** Let $\boldsymbol{R}_{\text{train}}^{(\tau)} = [\boldsymbol{r}_{\tau-w+1}, \cdots, \boldsymbol{r}_\tau] \in \mathbb{R}^{n \times w}$ denote the real log-return matrix for a split offset $\tau$ with length $w = 500 \times 24$ hours. A TSG model $\boldsymbol{g}$ is trained on $\boldsymbol{R}_{\text{train}}^{(\tau)}$ to capture both temporal dependencies and cross-sectional relationships. From this trained model, we sample synthetic returns:

$$\boldsymbol{R}_{\text{gen}} = \boldsymbol{g}(\boldsymbol{z}), \boldsymbol{z} \sim \mathcal{N}(\boldsymbol{0}, \boldsymbol{I}).$$

Next, features are extracted from $\boldsymbol{R}_{\text{gen}}$ via the pipeline: $\boldsymbol{\Phi}(\boldsymbol{R}_{\text{gen}}) \in \mathbb{R}^{n \times s \times d}$. A forecasting model $\boldsymbol{f} : \mathbb{R}^d \to \mathbb{R}$ then predicts the next-hour return:

$$\hat{r}_{i,t+1} = \boldsymbol{f}(\boldsymbol{\Phi}(\boldsymbol{R}_{\text{gen}})[i, t, :]).$$

We use XGBoost (Chen & Guestrin, 2016) as the forecasting model, chosen for its balance of robustness, interpretability, and minimal hyperparameter tuning (Vancsura et al., 2025; Liu et al., 2021; Yun et al., 2021), ensuring that benchmark results primarily reflect the quality of the generated data rather than model capacity.

**Trading Phase.** The trained forecaster is then applied to a test period of length $s = 30 \times 24$ hours. For each hour $t$ and asset $i$, we predict $\hat{r}_{i,t+1} = \boldsymbol{f}(\boldsymbol{\Phi}(\boldsymbol{R}_{\text{test}})[i, t, :])$, rank the vector $\hat{\boldsymbol{r}}_{t+1} = [\hat{r}_{i,t+1}]_{i=1}^n$, and construct a dollar-neutral portfolio by **longing** the top half of assets (highest $\hat{r}_{i,t+1}$) and **shorting** the bottom half (lowest $\hat{r}_{i,t+1}$). This portfolio is rebalanced hourly over the test window, maintaining balanced long and short exposures.

**Discussions.** This task reveals how well synthetic data generalizes to real markets, operationalizing the notion of functional realism. If $\boldsymbol{R}_{\text{gen}}$ preserves the predictive structures of $\boldsymbol{R}_{\text{train}}^{(\tau)}$, the realized P&L $\Delta V_t$ will score highly across CTBench's evaluation suite. Thus, synthetic data are valued not merely for statistical closeness to historical distributions but for the economic utility they unlock. Importantly, every component in Figure 3(a) is modular: researchers can substitute alternative TSG models, forecasters (e.g., Transformers), or feature sets, while retaining a unified scoring framework.

### C.2.2 STATISTICAL ARBITRAGE TASK

**Training Phase.** The Statistical Arbitrage task typically hinges on pairs or baskets of assets whose spreads revert toward a long-term mean. In this task, residuals between real $\boldsymbol{R}_{\text{train}}$ and reconstructed returns $\hat{\boldsymbol{R}}_{\text{train}}$ are assumed to follow mean-reverting dynamics. For asset $i$ and time $t$, we define training residual:

$$\rho_{i,t} = r_{i,t} - \hat{r}_{i,t},$$

where $r_{i,t} \in \boldsymbol{R}_{\text{train}}$ and $\hat{r}_{i,t} \in \hat{\boldsymbol{R}}_{\text{train}}$. For each asset $i$, these residuals are then fitted to an Ornstein–Uhlenbeck (OU) process (Uhlenbeck & Ornstein, 1930):

$$d\rho_{i,t} = \theta_i(\mu_i - \rho_{i,t})dt + \sigma_i dW_t,$$

where $\theta_i > 0$ (mean reversion speed), $\mu_i$ (long-run mean), and $\sigma_i$ (volatility) are estimated per asset, and $dW_t$ is a standard Wiener increment.

**Trading Phase.** On test data $\boldsymbol{R}_{\text{test}}$, the model reconstructs returns $\hat{\boldsymbol{R}}_{\text{test}}$, producing test residuals for $r_{i,t} \in \boldsymbol{R}_{\text{test}}$ and $\hat{r}_{i,t} \in \hat{\boldsymbol{R}}_{\text{test}}$:

$$\epsilon_{i,t} = r_{i,t} - \hat{r}_{i,t}.$$

Each residual $\epsilon_{i,t}$ is converted to an $s$-score:

$$s_{i,t} = \frac{\epsilon_{i,t} - \mu_i}{\sigma_i / \sqrt{2\theta_i}},$$

quantifying the deviation from equilibrium. Trading signals are then derived via:

- **Thresholding:** Open or maintain a short position if $s_{i,t} > +\gamma$, a long if $s_{i,t} < -\gamma$, otherwise stay flat, with $\gamma = 2$.
- **Weight Normalization:** Raw signals $\eta_{i,t} = \mathrm{sgn}(-s_{i,t}) \cdot |s_{i,t}|$ are normalized to $\tilde{\eta}_{i,t} = \eta_{i,t} / (\sum_j |\eta_{j,t}|)$.
- **Execution:** Portfolios are rebalanced hourly based on $\tilde{\eta}_{i,t}$.

**Discussions.** The Statistical Arbitrage task evaluates whether reconstructed time series reveal stable, mean-reverting residuals suitable for statistical arbitrage, complementing the generation-focused task by addressing market-neutral alpha extraction. These tasks ensure TSG models are tested not only for statistical fidelity but also for practical effectiveness in real-world crypto trading.

## C.3  TRADING STRATEGIES

Beyond the datasets and dual-task evaluation, trading strategies play a central role in assessing whether synthetic time series capture signals that are economically meaningful. Since no single strategy can fully characterize market behavior, CTBench incorporates multiple paradigms that reflect how practitioners deploy forecasts in real trading:

- **S1: Cross-Sectional Momentum (CSM)** takes long positions in the top decile and short positions in the bottom decile of assets ranked by predicted one-hour returns. This probes a model's ability to capture ranking-based alpha signals.
- **S2: Long-Only Top-Quantile (LOTQ)** equally weights and goes long in the top 20% of assets based on predicted returns, with all other weights set to zero. This isolates pure directional signals without short exposure.
- **S3: Proportional-Weighting (PW)** allocates weights proportionally to predicted returns: $\eta_{i,t} = \hat{r}_{i,t} / (\sum_{j=1}^{n} \hat{r}_{j,t})$, emphasizing the magnitude of forecasted signals rather than merely their ranks.

Each strategy exploits different statistical regularities, including level effects, cross-sectional dispersion, and serial correlations, ensuring that no single modeling flaw remains undetected. They span the primary mandates seen on crypto desks: beta-neutral long-short equity, directional trend capture, and volatility harvesting. Finally, the CTBench pipeline is fully **plug-and-play**. Traders can drop in any proprietary strategies without altering the benchmark code, fostering fair comparison across future studies.

## C.4  FINANCIAL METRIC SUITE

Evaluating TSG models in crypto requires more than statistical similarity; it demands metrics that connect directly to financial relevance and practical usability. To this end, CTBench employs a structured suite of measures that balance fidelity, predictive utility, trading performance, and risk management, while also accounting for computational efficiency.

**Error-based Evaluation.** Given the actual return $r_{i,t}$ and prediction $\hat{r}_{i,t}$ for asset $i$ and time $t$:

- **E1: Mean Squared Error (MSE)** is defined as:

$$\mathrm{MSE} = \frac{1}{k \cdot s \cdot n} \sum_{\tau \in \mathcal{O}} \sum_{t=1}^{s} \sum_{i=1}^{n} (r_{i,t+\tau} - \hat{r}_{i,t+\tau})^2.$$

- **E2: Mean Absolute Error (MAE)** is defined as

$$\mathrm{MAE} = \frac{1}{k \cdot s \cdot n} \sum_{\tau \in \mathcal{O}} \sum_{t=1}^{s} \sum_{i=1}^{n} |r_{i,t+\tau} - \hat{r}_{i,t+\tau}|.$$

Low values in both metrics reflect strong signal fidelity, while differences help distinguish outliers from widespread minor errors.

**Rank-based Evaluation.** Given realized returns $r_t$ and predictions $\hat{r}_t$ for all assets at time $t$:

- **E3: Information Coefficient (IC)** is defined as the average Spearman correlation between predicted and actual rankings, where $\text{IC}_{\tau,t} = \text{SpearmanCorr}(\boldsymbol{r}_{t+\tau}, \hat{\boldsymbol{r}}_{t+\tau})$. It is computed as:

$$\text{IC} = \frac{1}{k \cdot s} \sum_{\tau \in \mathcal{O}} \sum_{t=1}^{s} \text{IC}_{\tau,t}.$$

- **E4: Information Ratio (IR)** measures the stability of IC:

$$\text{IR} = \frac{\text{Mean}(\text{IC}_{\tau,t})}{\text{Std}(\text{IC}_{\tau,t})}.$$

A consistently positive IC shows the generator preserves rankings essential for long-short strategies, despite absolute errors.

**Trading Performance.** To assess economic utility, we evaluate both profitability and risk-adjusted returns using the hourly profit-and-loss $\Delta V_t$ and the simple return of equity $\Delta V_t/V_{t-1}$ at time $t$:

- **E5: Compound Annual Growth Rate (CAGR)** captures the annualized return based on equity growth, where $V_0$ and $V_s$ are the initial and final equity. It is calculated as:

$$\text{CAGR} = \left(\frac{V_s}{V_0}\right)^{8760/s} - 1.$$

- **E6: Sharpe Ratio (SR)** is defined as:

$$\text{SR} = \frac{\mathbb{E}[\Delta V_t/V_{t-1}]}{\text{Std}(\Delta V_t/V_{t-1})} \cdot \sqrt{8760}.$$

These metrics capture both returns and the risk profile of synthetic-data-driven strategies.

**Risk Assessment Metrics.** Given profit-and-loss series $\Delta V_t$ and simple return of equity $\Delta V_t/V_{t-1}$:

- **E7: Maximum Drawdown (MDD)** is defined as:

$$\text{MDD} = \max_{u \leq t} \left(\frac{V_u - V_t}{V_u}\right).$$

- **E8: Value at Risk (VaR)** at 95% confidence is defined as:

$$\text{VaR}_{0.95} = -\text{Percentile}_{5\%}(\Delta V_t/V_{t-1}).$$

- **E9: Expected Shortfall (ES)** at 95% confidence is defined as:

$$\text{ES}_{0.95} = -\mathbb{E}\big[(\Delta V_t/V_{t-1}) \mid (\Delta V_t/V_{t-1}) \leq -\text{VaR}_{0.95}\big].$$

VaR captures potential worst-day losses, while ES reveals mean loss beyond that threshold, offering a fuller picture of tail risk.

**Efficiency.** CTBench evaluates efficiency along two dimensions: training and inference time, to capture both scalability during model development and responsiveness in real-time use.

- **E10: Training Time** is the wall-clock time at which a TSG model is trained.

- **E11: Inference Time** is the mean wall-clock time to generate or reconstruct one batch of data ($n$ assets $\times$ $s$ time steps).

### C.5   TSG MODEL ZOO

**GAN-based Methods.** These methods (Seyfi et al., 2022; Wiese et al., 2020; Pei et al., 2021; Wang et al., 2023) leverage adversarial training to generate realistic series. They incorporate recurrent neural architectures and specialized attention mechanisms tailored to temporal dependencies.

- **M1: Quant-GAN (Wiese et al., 2020)** approximates a trading utility function, optimizing the generator for downstream profitability.
- **M2: COSCI-GAN (Seyfi et al., 2022)** integrates causal self-attention and statistical conditioning to consider temporal order and cross-asset correlations.

Table 3: Summary of popular TSG methods with their backbone models and financial datasets used.

| Year | Method | Backbone | Financial Datasets Used |
|------|--------|----------|------------------------|
| 2016 | C-RNN-GAN (Mogren, 2016) | GAN | / |
| 2017 | RCGAN (Esteban et al., 2017) | GAN | / |
| 2018 | T-CGAN (Ramponi et al., 2018) | GAN | / |
| 2019 | TimeGAN (Yoon et al., 2019) | GAN | Stocks |
| 2019 | WaveGAN (Donahue et al., 2019) | GAN | / |
| 2020 | COT-GAN (Xu et al., 2020) | GAN | / |
| 2020 | DoppelGANger (Lin et al., 2020) | GAN | / |
| 2020 | Quant-GAN (Wiese et al., 2020) | GAN | SPX |
| 2020 | SigCWGAN (Ni et al., 2020) | GAN | SPX & DJI |
| 2020 | TSGAN (Smith & Smith, 2020) | GAN | / |
| 2021 | RTSGAN (Pei et al., 2021) | GAN | Stocks |
| 2021 | Sig-WGAN (Ni et al., 2021) | GAN | SPX & DJI |
| 2021 | TimeGCI (Jarrett et al., 2021) | GAN | / |
| 2022 | CEGEN (Remlinger et al., 2022) | GAN | Stocks & Electric Price |
| 2022 | COSCI-GAN (Seyfi et al., 2022) | GAN | / |
| 2022 | PSA-GAN (Jeha et al., 2021) | GAN | / |
| 2022 | TsT-GAN (Srinivasan & Knottenbelt, 2022) | GAN | Stocks |
| 2022 | TTS-GAN (Li et al., 2022a) | GAN | / |
| 2023 | AEC-GAN (Wang et al., 2023) | GAN | / |
| 2023 | TT-AAE (Liu et al., 2023) | GAN | Stocks |
| 2021 | TimeVAE (Desai et al., 2021) | VAE | Stocks |
| 2023 | CRVAE (Li et al., 2023) | VAE | / |
| 2023 | TimeVQVAE (Lee et al., 2023) | VAE | / |
| 2024 | KoVAE (Naiman et al., 2024b) | VAE | Stocks |
| 2023 | DiffTime (Coletta et al., 2023) | Diffusion | Stocks |
| 2023 | TSGM (Lim et al., 2023) | Diffusion | Stocks |
| 2024 | Diffusion-TS (Yuan & Qiao, 2024) | Diffusion | Stocks |
| 2024 | FIDE (Galib et al., 2024) | Diffusion | Stocks |
| 2024 | ImagenTime (Naiman et al., 2024a) | Diffusion | Stocks |
| 2024 | SDformer (Chen et al., 2024) | Diffusion | Stocks |
| 2025 | PaD-TS (Li et al., 2025) | Diffusion | Stocks |
| 2020 | CTFP (Deng et al., 2020) | Flow | / |
| 2021 | Fourier-Flow (Alaa et al., 2021) | Flow | Stocks |
| 2024 | FlowTS (Hu et al., 2024) | Flow | Stocks |
| 2018 | Neural ODE (Chen et al., 2018) | ODE + RNN | / |
| 2019 | ODE-RNN (Rubanova et al., 2019) | ODE + RNN | / |
| 2021 | Neural SDE (Kidger et al., 2021) | ODE + GAN | Stocks |
| 2022 | GT-GAN (Jeon et al., 2022) | ODE + GAN | Stocks |
| 2023 | LS4 (Zhou et al., 2023) | ODE + VAE | / |
| 2024 | TimeLDM (Qian et al., 2024) | Diffusion + VAE | Stocks |

**VAE-based Methods.** These Methods use variational inference to capture both local and global temporal patterns (Desai et al., 2021; Lee et al., 2023; Li et al., 2023). They have shown strong performance in general TSG tasks (Ang et al., 2023a; Bao et al., 2024).

- **M3: TimeVAE (Desai et al., 2021)** is a sequence-aware VAE with temporal convolutions, designed to capture both local and long-range dependencies in multivariate time series.
- **M4: KoVAE (Naiman et al., 2024b)** enhances TimeVAE by incorporating Koopman operator-based latent dynamics for smoother and more interpretable generation.

**Diffusion-based Methods.** Diffusion models (Yuan & Qiao, 2024; Galib et al., 2024; Chen et al., 2024; Li et al., 2025; Naiman et al., 2024a) progressively convert noise into structured data via iterative denoising, proving highly effective in modeling complex market dynamics.

- **M5: Diffusion-TS (Yuan & Qiao, 2024)** is a score-based diffusion model that refines Gaussian noise into realistic trajectories, achieving state-of-the-art sample fidelity on financial data.
- **M6: FIDE (Galib et al., 2024)** introduces factorized conditional diffusion with attention-driven score networks, enabling conditional generation based on market regimes or liquidity factors.

**Flow-based Methods.** These methods (Alaa et al., 2021; Hu et al., 2024) employ invertible transformations to model data distributions, ensuring exact likelihood estimation and efficient sampling.

- **M7: Fourier-Flow (Alaa et al., 2021)** uses frequency-domain coupling layers for invertible transformations, allowing fast sampling and exact likelihood computation while preserving periodic structures.

**Mixed-based Methods.** Hybrid models (Zhou et al., 2023; Rubanova et al., 2019; Jeon et al., 2022) typically combine multiple modeling paradigms (e.g., ODEs and VAEs) to capture nuanced temporal dynamics and stochastic characteristics.

- **M8: LS4 (Zhou et al., 2023)** fuses deep state-space modeling with stochastic latent variables via variational inference, providing flexible and interpretable modeling of complex crypto dynamics.

# D  EXPERIMENTAL SETUP

**Datasets.** We employ the datasets (Binance Exchange, 2025b) described in **§3.1** for the experiments. To simulate real-world deployment, we adopt a walk-forward rolling-window validation scheme, using 500 days of hourly data for training, and 30 or 15 days for testing on the Predictive Utility and Statistical Arbitrage tasks, respectively. After each cycle, the window advances by the test period length, with models retrained. This process spans from January 2020 to December 2024, covering diverse market regimes.

**Benchmark Configurations.** To isolate core TSG model performance, we assume zero trading fees by default in both Predictive Utility and Statistical Arbitrage tasks, enabling fair comparison of signal quality without interference from platform-specific costs. For the Statistical Arbitrage task, we also apply a 0.03% trading fee, reflecting the fee level that a typical liquidity provider can achieve on major centralized exchanges (Zhang et al., 2023; Winkel & Härdle, 2023; Binance Exchange, 2025a), providing a more grounded evaluation of net profitability.

**Trading Strategies.** For the Predictive Utility task, we employ three representative trading strategies in **§3.3** to evaluate synthetic data across varied portfolio constructions. In contrast, the Statistical Arbitrage task employs the mean-reversion strategy to isolate the model's ability to preserve exploitable residual structures.

**TSG Methods.** We evaluate eight representative TSG models across five major families in **§3.5**. Hyperparameter settings follow published recommendations or are tuned for stable training.

- **GAN-based:** Quant-GAN adopts latent_dim = 8, hidden_dim = 80, gradient penalty $\lambda_{gp} = 10.0$, and critic steps $n_{critic} = 5$; COSCI-GAN uses latent_dim = 32, $\gamma = 5$, and $n_{groups} = 4$ with MLP-based central discriminators, as per (Seyfi et al., 2022).
- **VAE-based:** TimeVAE uses latent_dim = 8 with stacked hidden layers of 50, 100, and 200 units; KoVAE follows (Naiman et al., 2024b), setting $W_{KL} = 0.009$ and $W_{PRED} = 0.03$ for KL and auxiliary loss terms.
- **Diffusion-based:** Diffusion-TS uses 1000 timesteps, 3 encoder layers, 6 decoder layers, and $d_{model} = 64$; FIDE applies 1000 steps, hidden_dim = 64, 8 layers, and $\sigma = 0.05$.
- **Flow-based:** Fourier-Flow incorporates DFT-based coupling layers with hidden_size = 128 and 3 flow layers.
- **Mixed-type:** LS4 employs hidden_dim = 6, latent_dim = 8, and a batch size of 512.

**Financial Baseline Models.** We added two baselines commonly used in quantitative finance:

- **ARMA-GARCH** (Engle, 1982): We fit multivariate VAR-DCC-GARCH models, which are multivariate extensions of univariate ARMA-GARCH models, to jointly capture the conditional

Figure 14: Simulated growth curves of $10,000 over four years under three trading strategies.

mean, volatility, and dynamic correlation structure of returns. Specifically, we use a VAR(1) specification for the conditional mean (corresponding to an ARMA(1,0) structure in each marginal series), univariate GARCH(1,1) processes for the conditional variances, and a DCC(1,1) structure for the time-varying conditional correlations.

- **Bootstrap** (Rubin, 1981): We implement a bootstrap-style generator that resamples returns to preserve marginal distributions and local dependence structures without explicit parametric dynamics.

These models are evaluated under the same Predictive Utility task as the learning-based TSG models. In addition, for the Statistical Arbitrage task, we use the ARMA-GARCH model's filtered (fitted) conditional mean process as reconstructions as in **§3.2**.

**Evaluation Metrics.** We adopt the thirteen metrics detailed in **§3.4**, thereby scoring each model on forecasting accuracy, rank correlation, trading profitability, tail risk, and computational efficiency.

**Experiments Environments.** All experiments are conducted on a machine equipped with an Intel® Xeon® Platinum 8480C @3.80GHz, 64 GB RAM, and an NVIDIA H100 GPU.

# E    ADDITIONAL RESULTS ON EQUITY CURVE DYNAMICS

## E.1    PREDICTIVE UTILITY TASK

Figure 14 presents log-scaled equity curves (initialized at $10,000) for each TSG model under three trading strategies from 2021 to 2024, highlighting cumulative returns and how inductive biases interact with shifting market regimes.

Under **CSM**, COSCI-GAN and TimeVAE deliver steady gains by effectively preserving rank order and alpha signals, though both limit upside potential by dampening extreme winners. In contrast, Diffusion-TS and FIDE exhibit persistent declines, as their denoising processes suppress volatility and weaken long-short execution.

Under **LOTQ**, COSCI-GAN clearly outperforms, likely benefiting from adversarially enhanced right-tail signals that capture strong directional momentum. TimeVAE and Fourier-Flow sustain modest but stable growth, whereas Diffusion-TS falters by missing rare yet critical upward spikes.

Finally, under **PW**, which emphasizes consistent pairwise rankings, COSCI-GAN again dominates. TimeVAE and Fourier-Flow show smooth compounding, reflecting robust generalization from well-regularized latent spaces. By contrast, LS4 remains largely flat across all strategies, suggesting its conservative design behaves more like a low-beta portfolio than an alpha-generating model.

## E.2    STATISTICAL ARBITRAGE TASK

Figure 15 illustrates the equity curves under the Statistical Arbitrage task, starting from $10,000 and incorporating 0.03% trading fees.

At the top end, LS4 compounds almost monotonically, underscoring its strong fee resilience, with two staircase-like surges in mid-2022 and early 2023. This pattern suggests that its latent-switching mechanism is particularly effective at capturing regime shifts rather than merely reacting to incremental mean-reversion. KoVAE follows with a similarly convex trajectory, initially smooth with shallow drawdowns through 2023, though its growth fades amid the turbulence of 2024. TimeVAE maintains steady gains through 2022, plateaus in mid-2023, and drifts sideways or slightly down-

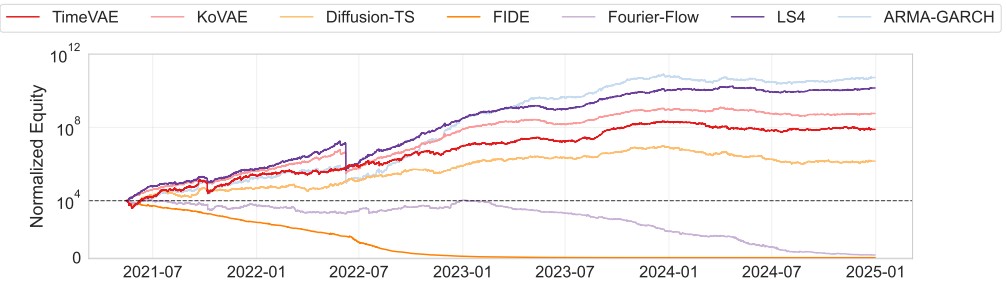

Figure 15: Simulated growth curves of $10,000 for the Statistical Arbitrage task (with 0.03% fee).

ward in 2024, reflecting its dependence on residual signals that weaken as cross-sectional dispersion compresses.

Among mid-tier performers, Diffusion-TS delivers a notably stable curve with minimal drawdowns, though its terminal return is the lowest among viable models, consistent with its characterization as a fee-resilient, risk-balanced generator.

At the lower end, FIDE collapses early, suggesting that its residuals may be over-regularized to the point of eliminating tradable structure. Meanwhile, Fourier-Flow experiences a slow but persistent bleed after mid-2022, likely driven by over-smoothed residual patterns that generate a sustained negative carry.

# F   THE USE OF LARGE LANGUAGE MODELS (LLMS)

LLMs are used mainly as auxiliary tools to aid and polish the writing of the paper.

