# OpenReview forum: "CTBench: Cryptocurrency Time Series Generation Benchmark"
_ICLR.cc/2026/Conference — ICLR 2026 Poster_

### Official Review · Reviewer_WGHC · 2025-10-28

**Soundness:** 2
**Presentation:** 3
**Contribution:** 3
**Rating:** 4
**Confidence:** 3

**Summary:**

CTBench is the benchmark tailored for Time Series Generation  in cryptocurrency markets, addressing limitations of existing benchmarks in domain generality, task scope, and crypto-specific evaluation. It features a curated dataset of OHLC data, dual-task evaluation, 13 financial metrics across 5 dimensions, and 8 TSG models from 5 families. Experiments across bull, volatile, consolidation, and mean-reverting regimes reveal trade-offs between statistical fidelity and trading profitability. CTBench enables rigorous TSG evaluation for crypto trading, bridging synthetic data generation with practical financial utility.

**Strengths:**

1. Captures unique traits of crypto markets including 24/7 volatility, absence of intrinsic valuation and irregular liquidity that are not addressed in traditional benchmarks.
2. Links TSG to real-world use cases through two complementary tasks focusing on forecasting capabilities and tradable signal extraction rather than just statistical similarity.
3. Integrates 13 diverse metrics covering error measurement, rank correlation, trading performance, risk assessment and computational efficiency for comprehensive evaluation of synthetic data’s financial utility.
4. Benchmarks 8 representative TSG models spanning five architectural families across four distinct market regimes, delivering actionable scenario-specific recommendations for practitioners.

**Weaknesses:**

1. Relies solely on Binance’s spot hourly data, lacking the diversity of cross-exchange data, alternative crypto asset types such as futures contracts, and different sampling frequencies. Besides, it remains unknown whether this generation method can be effectively extended to a broader range of asset classes, which limits the method’s scalability.
2. All TSG models' generated data are used with the same XGBoost model for downstream prediction. Different TSG models may be better suited for different types of predictors as some models like Transformer are more sensitive to long-term dependencies, so using a fixed predictor may introduce evaluation bias and fail to fully reflect the generality of the generated data.
3. Diffusion-based models included in the benchmark may exhibit long training and inference times, providing comparisons of generation efficiency across different models would help alleviate concerns in this regard.

**Questions:**

See Weakness.

---

> ### Author Response · Authors · 2025-12-04
> **Official Comment by Authors (1/2)**
>
> We thank the reviewer for the careful reading and for highlighting both the crypto-specific strengths of CTBench and the practical value of our dual-task evaluation and financial-metric design.
> We respond to each point below.
>
> > W1. Relies solely on Binance's spot hourly data, lacking the diversity of cross-exchange data, alternative crypto asset types such as futures contracts, and different sampling frequencies. Besides, it remains unknown whether this generation method can be effectively extended to a broader range of asset classes, which limits the method's scalability.
>
> We appreciate this important comment and would like to clarify a key point upfront: **CTBench is a benchmark, not a generation method**.
>
> The dual-task pipeline in Section 3 is **explicitly model-agnostic**: any TSG method that outputs a multivariate return series can be plugged in. Our goal is to standardize **data, tasks, and evaluation**, not to propose a specific generative model that must scale to all markets.
>
> Importantly, **nothing in the benchmark is specific to Binance, spot trading, or hourly frequency**. Any return panel, across exchanges, assets, or frequencies, could be integrated as an additional dataset module.
>
> We chose **Binance spot USDT pairs** for CTBench for three practical and scientific reasons, which we will make explicit in **Appendix B**:
>
> **(1) Market Representativeness**
>
> Multiple independent industry sources confirm that Binance is the **largest centralized exchange by global spot volume** (35~47% in 2024--2025) [1,2,3], and Binance is also among the **largest derivatives exchanges** by 24h trading volume and open interest [4,5].
> Aggregator sites such as CoinGecko [6] and CoinDesk [7] similarly list Binance as the top CEX by volume and number of markets. Focusing on Binance spot thus **covers the most liquid and widely used venue** in crypto markets, capturing a large fraction of real trading activity.
>
> Moreover, nearly all major Binance pairs are also listed elsewhere, meaning these prices are informative for the broader market while avoiding the confounding effects of cross-exchange differences in fee structures, matching engines, and data formats.
>
> **(2) Data Quality, Coverage, and Accessibility**
>
> Our curated dataset (in Section 3.1) contains **452 USDT pairs** with continuous hourly data from 2020--2024, covering large-cap, mid-cap, altcoins, and DeFi assets. Hourly frequency offers a strong balance of:
> - reduced microstructure noise,
> - alignment with common systematic trading horizons,
> - tractable panel size for 5-year coverage with 452 assets.
>
> Crucially, Binance provides this entire history **publicly and for free**. Comparable datasets from other major exchanges (e.g., Coinbase [8]) typically require **institutional paid access**, which would make CTBench **non-reproducible** for many academic users. Ensuring **open reproducibility** is a core design principle of CTBench.
>
> **(3) Benchmark Focus and Extensibility**
>
> CTBench provides a clean, representative, reproducible core for crypto TSG. The entire design is modular: other exchanges, futures, or sampling intervals can be added as new dataset modules without modifying the task or metric suite.
> We will clarify this extensibility in Appendix B and discuss multi-exchange and multi-frequency extensions as natural next steps.
>
> Finally, regarding **derivatives / perpetual futures**: In crypto, perpetual contracts require modeling **funding fees**, which periodically transfer value between long and short positions and affect realized PnL independently of return prediction. Including them in CTBench would require specifying explicit assumptions on funding-payment accounting and position management.
> To keep CTBench focused and clean, we chose spot markets; a futures-focused extension, with explicit funding-rate handling, is planned for future versions.
>
> **In summary**, CTBench deliberately targets the largest, cleanest, most accessible portion of crypto markets, while its architecture is fully extensible to broader asset classes and data sources.

---

> > ### Author Response · Authors · 2025-12-04
> > **Official Comment by Authors (2/2)**
> >
> > > W2. All TSG models' generated data are used with the same XGBoost model for downstream prediction. Different TSG models may be better suited for different types of predictors as some models like Transformer are more sensitive to long-term dependencies, so using a fixed predictor may introduce evaluation bias and fail to fully reflect the generality of the generated data.
> >
> > Thank you for raising this concern.
> > To directly test the sensitivity of CTBench's conclusions to the choice of predictor, we conducted a **comprehensive ablation over five forecasting architectures: Linear Regression, Random Forest, MLP, Transformer, and XGBoost**, all trained on synthetic data from every TSG model and evaluated on real returns using the same walk-forward protocol. Full results are now reported in **Section 4.4 (Ablation Study)**.
> >
> > Across all experiments, three robust patterns emerge:
> > - **Linear Regression and MLP** produce large MSE/MAE and near-zero IC/IR, showing weak predictive and cross-sectional ability.
> > - **Random Forest and Transformer** reduce prediction errors but retain near-zero IC/IR, capturing short-term noise rather than tradable structure.
> > - **XGBoost** consistently provides the strongest combination of low error and high rank correlation, making it the *most discriminative* and *trading-relevant* forecaster.
> >
> > These findings demonstrate that **XGBoost is the most reliable evaluator of tradable structure**, justifying its use as the default forecaster in CTBench.
> >
> >
> > > W3. Diffusion-based models included in the benchmark may exhibit long training and inference times, providing comparisons of generation efficiency across different models would help alleviate concerns in this regard.
> >
> > We agree that efficiency is critical, especially for frequently retrained crypto systems.
> > CTBench already includes such comparisons: **Section 4.3 and Figure 9** report wall-clock **training and inference times** for all eight TSG models, including both diffusion models (Diffusion-TS, FIDE).
> > We also explicitly discuss the **accuracy-efficiency trade-off**, showing that diffusion models yield high fidelity but are better suited for offline pipelines due to their longer runtimes.
> >
> > We will emphasize this more clearly in the revision to ensure readers do not overlook the included efficiency analysis.
> >
> > **Summary of Revisions**
> >
> > In response to the reviewer, we have:
> > - Clarified that **CTBench is model-agnostic** and justified the dataset choice based on market representativeness, data quality, accessibility, and reproducibility.
> > - Conducted a **multi-forecaster ablation**, confirming that CTBench's comparative insights are robust across diverse predictors.
> > - Highlighted existing **efficiency evaluations** and improved clarity around runtime comparisons, especially for diffusion models.
> > - Improved the paper by adding explanations on extensibility and clarifying the scope of CTBench.
> >
> > We thank the reviewer again for the thoughtful and constructive feedback and believe these revisions greatly strengthen the clarity, scalability, and practical relevance of CTBench.
> >
> >
> > **Reference**
> >
> > [1] https://www.binance.com/en-BH/square/post/18929557100097/
> >
> > [2] https://www.coingecko.com/research/publications/centralized-crypto-exchanges-market-share
> >
> > [3] https://www.financemagnates.com/cryptocurrency/global-spot-crypto-trading-climbs-142-year-over-year-to-21t/
> >
> > [4] https://www.binance.com/en/futures/home
> >
> > [5] https://coinmarketcap.com/rankings/exchanges/derivatives/
> >
> > [6] https://www.coingecko.com/en/exchanges
> >
> > [7] https://data.coindesk.com/reports/exchange-review-march-2023
> >
> > [8] https://help.coinbase.com/en/data-marketplace/getting-started/coinbase-data-marketplace

---

### Official Review · Reviewer_8b4R · 2025-10-30

**Soundness:** 3
**Presentation:** 3
**Contribution:** 3
**Rating:** 8
**Confidence:** 4

**Summary:**

This paper provides an open-source cryptocurrency dataset and the first benchmark for time series generation in cryptocurrency, named CTBench. Moreover, it introduces a novel dual-task evaluation framework to assess how well the temporal dynamics captured by synthetic data can be used for downstream forecasting and whether they can reconstruct market structure and isolate tradable signals. CTBench specifically evaluates the capability of eight state-of-the-art generative models for time series, using abundant metrics in different financial applications. Experimental results benchmark the performance of synthetic data from tested generative models.

**Strengths:**

- This paper offers a valuable open-source cryptocurrency dataset, which bridges the gap between research on time series generation and cryptocurrency.

- The benchmark discussed in this paper is comprehensive, involving most of recent deep generative models and evaluation metrics widely used in finance.

- This paper provides some intriguing practical findings that are critical for future research and applications: (i) current generative models achieve diverse performance across tasks, none of them can outperform others across all tasks and metrics; (ii) in statistical arbitrage task, different generative models are sensitive to distinct scenarios; (iii) noise in financial data is important as well.

- Beyond the above performance, this paper further evaluates the efficiency of generative models.

- The writing of this paper is well-structured and easy to read.

**Weaknesses:**

- The predictive utility task fixes the forecasting model to XGBoost for its robustness and interpretability. However, some conclusions may change with alternative predictors, given that different setups have large impacts on the model performance.

- [Minor] Dual-task evaluation module in Figure 2 is visually unclear. Perhaps its readability can be improved by simplifying the contents like the other modules, since Figure 3 already shows the details.

**Questions:**

- Is it possible that generative models may have different performance using forecasting models other than XGBoost?

- Are all feature windows strictly backward-looking and computed within the split? Are any scalers fit only on training data? Do all the results hold using different splits?

- A small typo: Citation formatting at line 041 is incorrect.

---

> ### Author Response · Authors · 2025-12-04
>
> We thank the reviewer for the positive and detailed assessment, and we appreciate the constructive suggestions.
> We are glad that you find the dataset, benchmark design, and empirical findings valuable for advancing research and applications in financial time-series generation.
> Below we address each concern in turn.
>
> > W1. The predictive utility task fixes the forecasting model to XGBoost for its robustness and interpretability. However, some conclusions may change with alternative predictors, given that different setups have large impacts on the model performance.
>
> > Q1. Is it possible that generative models may have different performance using forecasting models other than XGBoost?
>
> We appreciate this important observation.
> To directly assess whether our conclusions depend on the choice of forecaster, we conducted a **comprehensive ablation across five forecasting architectures: Linear Regression, Random Forest, MLP, Transformer, and XGBoost**, all trained on synthetic data from every TSG model and evaluated on real returns using the same walk-forward protocol. Full results are now reported in **Section 4.4 (Ablation Study)**.
>
> Across all experiments, we observe three consistent patterns:
> - **Linear Regression and MLP** exhibit high MSE/MAE and near-zero IC/IR, indicating limited ability to extract either pointwise or cross-sectional signals from synthetic data.
> - **Random Forest and Transformer** reduce prediction errors but still fail to capture rank structure, yielding weak IC/IR.
> - **XGBoost** consistently achieves the best balance of low error and strong rank correlation, providing the clearest differentiation among TSG models.
>
> These findings confirm that **XGBoost offers the most stable and discriminative evaluation signal for tradable structure**, validating its role as the default forecasting model in CTBench.
>
>
> > Q2. Are all feature windows strictly backward-looking and computed within the split? Are any scalers fit only on training data? Do all the results hold using different splits?
>
> We appreciate the opportunity to clarify these design choices:
> 1. **Feature computation is strictly backward-looking**.
> All features, including technical indicators, are computed within each rolling-window split using *only past returns*. No look-ahead leakage occurs.
>
> 2. **Scaling follows standard practice**.
> When normalization is applied, the scaler is fit exclusively on training features and then applied to the corresponding test features within each split.
>
> 3. **Results are robust to split configuration**.
> Our evaluation averages over multiple walk-forward splits spanning four distinct market regimes (2021--2024) under a 500-day / 30-day train-test structure, which aligns with real-world crypto backtesting standards. We found that relative model rankings remain stable across these regimes.
>
>
> > W2. [Minor] Dual-task evaluation module in Figure 2 is visually unclear. Perhaps its readability can be improved by simplifying the contents like the other modules, since Figure 3 already shows the details.
>
> > Q3. A small typo: Citation formatting at line 041 is incorrect.
>
> Thank you for the helpful suggestions.
> We have simplified the dual-task module in **Figure 2** to match the visual clarity of the other blocks, and we direct readers to **Figure 3** for detailed task mechanics.
> We have also corrected the citation formatting at line 041 and reviewed the manuscript for additional consistency issues.
>
> **Summary of Revisions**
>
> In response to your feedback, we have:
> - Added a **multi-forecaster ablation**, demonstrating that CTBench's comparative findings are robust across different predictive models.
> - Clarified the **backward-looking feature construction**, **training-only scaling**, and **walk-forward split robustness**.
> - Improved the **readability of Figure 2** and corrected the reported citation issue.
>
> We appreciate your encouraging evaluation and believe these revisions further strengthen the clarity, rigor, and usability of CTBench.

---

### Official Review · Reviewer_hL6x · 2025-10-31

**Soundness:** 3
**Presentation:** 3
**Contribution:** 3
**Rating:** 6
**Confidence:** 4

**Summary:**

The paper introduces CTBench, an open-source benchmark for crypto time‑series generation. It provides a curated, hourly panel of 452 USDT pairs on Binance (2020–2024) and a two-track evaluation for crypto time-series generation. Representative generators are tested across four market regimes using a finance-first metric stack that spans several aspects. The results document model utility in different regimes.

**Strengths:**

- Empirically, the authors observe the regime-dependent trade-offs between fidelity and tractability, i.e., high fidelity does not directly imply tradability. This is an important insight for generating synthetic financial data.

- Emphasizing rank-based metrics and arbitrage capacity over purely generative scores moves the discussion toward decision-useful validation.

- The authors offer useful guidance on model selection for practical uses.

**Weaknesses:**

- Results are tied to a single forecasting model. Without testing a variety of forecasters, it’s unclear whether conclusions generalize.

- Insufficient diagnostic analysis of model performance differentials. The paper does not investigate why the TSG models diverge in outcomes, leaving the observations unexplained.

**Questions:**

- Do you evaluate the covariance and eigenspectrum alignment between synthetic and real returns? This might explain why Fourier-Flow yields stable rank metrics but limited arbitrage.

- Compared with those used in finance economics, such as ARMA-GARCH and the bootstrap, do the learning-based TSG models provide better synthetic data?

- Have you tested alternative forecasters (e.g., linear model, random forest, and MLP)? Do the relative model performance and rankings persist?

---

> ### Author Response · Authors · 2025-12-04
>
> We sincerely thank the reviewer for the careful reading and thoughtful comments.
> We are encouraged that you found CTBench's empirical insights, dual-task design, and finance-first evaluation perspective valuable.
> We have carefully addressed all concerns, added new analyses, and incorporated the corresponding revisions in the updated manuscript. Below, we respond to each point in detail.
>
> > W1. Results are tied to a single forecasting model. Without testing a variety of forecasters, it's unclear whether conclusions generalize.
>
> > Q3. Have you tested alternative forecasters (e.g., linear model, random forest, and MLP)? Do the relative model performance and rankings persist?
>
> We appreciate this important observation.
> To directly address potential evaluator bias, we conducted a comprehensive ablation over **five forecasting architectures: Linear Regression, Random Forest, MLP, Transformer, and XGBoost**, all trained on synthetic data from every TSG model and evaluated on real returns using the same walk-forward protocol. Full results are now reported in **Section 4.4 (Ablation Study)**.
>
> Across all experiments, three robust patterns emerge:
> - **Linear Regression and MLP** produce large MSE/MAE and near-zero IC/IR, showing weak predictive and cross-sectional ability.
> - **Random Forest and Transformer** reduce prediction errors but retain near-zero IC/IR, capturing short-term noise rather than tradable structure.
> - **XGBoost** consistently provides the strongest combination of low error and high rank correlation, making it the *most discriminative* and *trading-relevant* forecaster.
>
> These findings confirm that **XGBoost is the most reliable evaluator of tradable structure**, justifying its use as the default forecaster in CTBench.
>
>
> > Q2. Compared with those used in finance economics, such as ARMA-GARCH and the bootstrap, do the learning-based TSG models provide better synthetic data?
>
> We appreciate the suggestion to compare CTBench's TSG models with classical econometric approaches. In the revised manuscript, we add two widely used baselines:
> - **ARMA-GARCH (VAR-DCC-GARCH implementation) [1]:** Captures conditional mean, volatility clustering, and dynamic correlation structure of returns. We use a VAR(1) specification for the conditional mean (corresponding to an ARMA(1,0) structure in each marginal series), univariate GARCH(1,1) processes for the conditional variances, and a DCC(1,1) structure for the time-varying conditional correlations.
> - **Bootstrap (Bayesian bootstrap) [2]:** Resamples real returns to preserve marginal distribution and local dependence.
>
> On the **Predictive Utility** task:
> - ARMA-GARCH and Bootstrap underperform on MSE/MAE, IC/IR, Sharpe, and CAGR.
> - ARMA-GARCH provides conservative but low-alpha signals.
> - Bootstrap produces moderate fidelity but weak tradability.
> - Learning-based TSG models significantly outperform them in rank metrics and trading performance, capturing cross-sectional alpha and regime shifts that classical models miss.
>
> On the **Statistical Arbitrage** task (ARMA-GARCH only; Bootstrap cannot reconstruct inputs):
> - ARMA-GARCH achieves a reasonable CAGR but suffers from poor VaR/ES due to large tail losses.
> - TSG models provide more balanced risk-return profiles.
>
> These comparisons are now integrated into **Sections 4.1 and 4.2**, respectively.
>
>
> > W2. Insufficient diagnostic analysis of model performance differentials. The paper does not investigate why the TSG models diverge in outcomes, leaving the observations unexplained.
>
> > Q1. Do you evaluate the covariance and eigenspectrum alignment between synthetic and real returns? This might explain why Fourier-Flow yields stable rank metrics but limited arbitrage.
>
> We agree that deeper structural diagnostics strengthen the interpretability of CTBench. We now include **covariance and eigenspectrum analysis** for all TSG models, with new findings that clarify the reviewer's question about Fourier-Flow:
>
> **Key Insight:**
> - Fourier-Flow's weak arbitrage performance **does not** stem from poor covariance modeling.
> - Due to the **invertibility of normalizing flows**, Fourier-Flow reconstructs returns with *near-lossless* precision: residual errors ~$10^{-5}$ and eigenspectrum deviation ~$10^{-9}$.
>
> Paradoxically:
> - **Perfect reconstruction leaves no mean-reverting spreads**, producing only numerical noise.
> - Therefore, **Predictive Utility remains strong** (rank structure preserved), while **Statistical Arbitrage fails** (no residual variance to trade on).
>
> This diagnostic has been added to the revised manuscript and explicitly explains the divergence between Fidelity and Arbitrage for flow-based models.
>
> **Reference**
>
> [1] Robert F Engle. Autoregressive conditional heteroscedasticity with estimates of the variance of united kingdom inflation. Econometrica: Journal of the Econometric Society, pp. 987-1007, 1982.
>
> [2] Donald B Rubin. The bayesian bootstrap. The Annals of Statistics, pp. 130-134, 1981.

---

> > ### Author Response · Authors · 2025-12-04
> >
> > **Summary of Revisions**
> >
> > To comprehensively address reviewer concerns, we added:
> > - **Ablation over five forecasters**, confirming XGBoost as the most discriminative evaluator.
> > - **Two classical baselines (ARMA-GARCH and Bootstrap)** and detailed comparisons on both tasks.
> > - **Covariance and eigenspectrum diagnostics**, explaining model-specific behaviors such as Fourier-Flow's mismatch between rank fidelity and arbitrage utility.
> >
> > Together, these additions significantly strengthen CTBench's robustness, interpretability, and practical relevance.
> >
> > We hope these revisions satisfactorily address all concerns and further improve the clarity and scientific value of the benchmark.

---

### Author Response · Authors · 2025-12-04
**Revision Summary for AC**

Dear AC,

Thank you for coordinating the review process and for the constructive feedback from Reviewers **hL6x**, **8b4R**, and **WGHC**.
We are encouraged that all three reviewers highlighted CTBench's strengths: its crypto-centric dataset, dual-task evaluation (predictive utility + statistical arbitrage), and comprehensive financial metric suite, and regarded the benchmark as timely and valuable for the TSG community.

Below, we summarize the key concerns raised and the corresponding revisions, additions, and clarifications made in the updated manuscript.

>**1. Dependence on a Single Forecaster (hL6x Q3; 8b4R W1/Q1; WGHC W2)**

All reviewers asked whether the findings depend on using **XGBoost** in the Predictive Utility task.

To address this, we added a **new ablation study (Section 4.4, Figure 10)** comparing five forecasting architectures: Linear Regression, Random Forest (RF), MLP, Transformer, and XGBoost.

Across all TSG models and time periods, **XGBoost consistently provides the strongest combination of low error and high IC/IR**, while other forecasters either underfit (Linear/MLP) or capture short-term noise rather than tradable structure (RF/Transformer).
This confirms that **CTBench's conclusions are robust** and that XGBoost is an appropriate default evaluator of tradable structure.

>**2. Comparison with Classical Finance Baselines (hL6x Q2)**

Reviewer hL6x asked how learning-based TSG compare with traditional econometric approaches.
We added two widely used baselines:
- **ARMA-GARCH** (VAR-DCC-GARCH): Evaluated on both Predictive Utility and Statistical Arbitrage.
- **Bootstrap**: Evaluated on Predictive Utility (non-predictive by construction).

**Findings:**
- Classical models can be competitive on error metrics but struggle on **rank-based and trading metrics**, especially when cross-sectional dispersion or regime shifts matter.
- Learning-based TSG models, particularly **TimeVAE, COSCI-GAN, and KoVAE**, better preserve alpha structure and deliver **higher Sharpe/CAGR with comparable or lower tail risk**.

These results help position CTBench as a bridge between traditional financial modeling and modern TSG methods.

>**3. Benchmark Scope and Extensibility (WGHC W1)**

We clarified that CTBench is a **model-agnostic, dataset-agnostic benchmark**, not a generation method.
The data loader, feature extraction, and evaluation pipeline are **fully modular** and can incorporate:
- multi-exchange spot data,
- futures or derivatives data (with funding-rate extensions),
- alternative sampling frequencies.

Binance hourly USDT pairs were selected for **liquidity, data quality, and public reproducibility**, not as a limitation. We expanded Appendix B to explicitly explain this design choice and the benchmark's extensibility.

> **4. Clarification and Presentation Improvements (8b4R W2/Q2/Q3)**

Reviewer 8b4R requested clarifications on feature engineering, scaling, and figure readability.

In the revision:
- All features are confirmed to be **strictly backward-looking**,
- All scalers are fit **only on training splits**,
- The walk-forward procedure is fully described.
- The dual-task module in Fig. 2 is simplified for readability, and all citation formatting issues are corrected.

**Summary**

In response to reviewer feedback, we have:
- Added a **multi-forecaster ablation**, confirming robustness of CTBench results.
- Added **ARMA-GARCH and Bootstrap** as classical baselines.
- Clarified benchmark **scope, extensibility, and data-choice rationale**.
- Improved **figures, formatting, and methodological descriptions** to ensure clarity and reproducibility.

We believe these revisions substantially strengthen the manuscript and directly address the reviewers' concerns.
We appreciate your consideration and hope these improvements will be helpful during the discussion and final decision process.

---

### Meta-Review · Area_Chair_5Vjg · 2025-12-13

**Summary:**

This paper has a mixture of review scores, with reasonable variance.  It provides a valuable benchmark for cryptocurrency time series, and the dual-task evaluation framework is innovative and useful in a more comprehensive benchmark. The major concerns are about the selection of the scenario and granularity and the class of algorithms evaluated. Most concerns are addressed or explained in the rebuttal phase. So I think it could be a solid paper in ICLR, and will be a helpful benchmark for future research.

**Reviewer Concerns:**

More concerns are addressed or explained.

**Reviewer Scores:**

The score from Reviewer WGHC could be increased. Major concerns from him/her are addressed.

---

### Decision · Program_Chairs · 2026-01-26

Accept (Poster)